# BodyGen: Advancing Towards Efficient Embodiment Co-Design

**Haofei Lu[1], Zhe Wu[1], Junliang Xing[*1], Jianshu Li[2], Ruoyu Li[2], Zhe Li[2], Yuanchun Shi[1]**
[1]Department of Computer Science and Technology, Tsinghua University, [2]Ant Group
{luhf23,wu-z24}@mails.tsinghua.edu.cn, {jlxing,shiyc}@tsinghua.edu.cn
{jianshu.l,ruoyu.li,lizhe.lz}@antgroup.com

## ABSTRACT

Embodiment co-design aims to optimize a robot's morphology and control policy simultaneously. While prior work has demonstrated its potential for generating environment-adaptive robots, this field still faces persistent challenges in optimization efficiency due to the (i) combinatorial nature of morphological search spaces and (ii) intricate dependencies between morphology and control. We prove that the ineffective morphology representation and unbalanced reward signals between the design and control stages are key obstacles to efficiency. To advance towards efficient embodiment co-design, we propose *BodyGen*, which utilizes (1) topology-aware self-attention for both design and control, enabling efficient morphology representation with lightweight model sizes; (2) a temporal credit assignment mechanism that ensures balanced reward signals for optimization. With our findings, Body achieves an average **60.03%** performance improvement against state-of-the-art baselines. We provide codes and more results on the website: https://genesisorigin.github.io.

## 1 INTRODUCTION

Species in nature are blessed with millions of years to evolve for remarkable capacities to adapt to the environment (Pfeifer & Scheier, 2001; Vargas et al., 2014). Time has gifted them with perfect physical bodies for movement and navigation, powerful processors for centralized information processing, and effective actuators for rapid interaction with their surroundings. Inspired by this observation, *embodiment co-design* (Sims, 1994; Ha, 2019; Yuan et al., 2021; Wang et al., 2023), where a robot's morphology and control policy are optimized

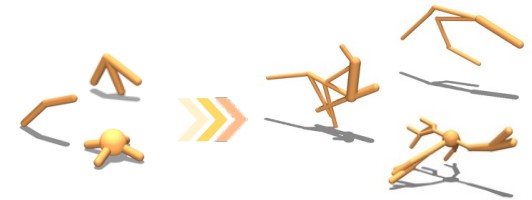

(a) Initial Designs      (b) Morphologies with Policies

Figure 1: Embodied Agents generated by BodyGen.

simultaneously, has gained increasing attention and demonstrates significant potential in various downstream fields, such as automated robot design and bio-inspired robot generation (Kriegman et al., 2020; Nakajima et al., 2018; Judd et al., 2019; Pan et al., 2021; Whitman et al., 2023). However, this task encounters extreme difficulties: (1) the morphology search space is quite vast and combinatorial, with each morphology corresponding to unique action and state spaces; (2) evaluating each candidate design requires an expensive roll-out to find its optimal control policy, which is almost unfeasible for the expensive computation.

Traditional evolutionary strategies (Sims, 1994; Wang et al., 2018b; Zhao et al., 2020; Gupta et al., 2021b) address these challenges through mutation-based population optimization but suffer from inefficient sampling and scalability limitations, requiring large amounts of computation. While structural constraints like symmetry (Gupta et al., 2021b; Dong et al., 2023) reduce constant search complexity, such human priors may compromise functionality (Yuan et al., 2021). Alternative

---

[*]Corresponding Author

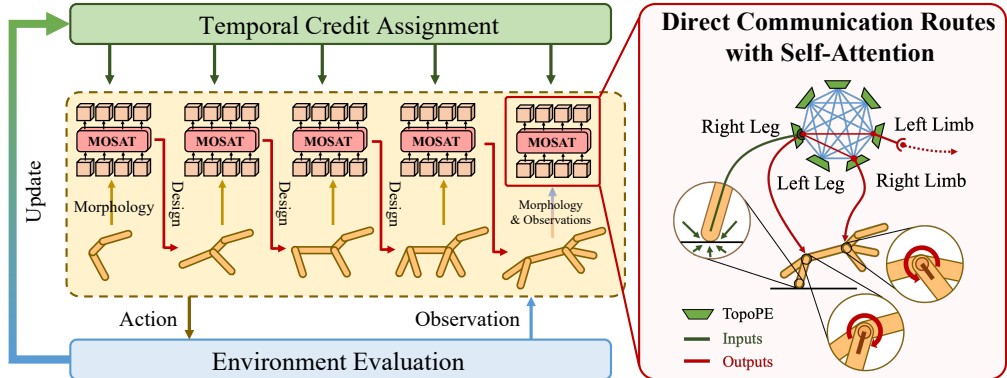

Figure 2: Overview of BodyGen, which leverages an RL-based framework for joint evolving of morphology and control policy, and an attention-based network equipped with Topology Position Encoding (TopoPE) for centralized message processing.

approaches employ modular GNN-based controllers (Huang et al., 2020; Yuan et al., 2021) for cross-morphology policy sharing, yet struggle with effective joint-level message aggregation (Kurin et al., 2020).

In this work, we aim to further push the potential of embodiment co-design, by introducing a method that enables efficient generation of high-performance embodied agents while maintaining computational affordability. Here, we announce BodyGen, a reinforcement learning framework for efficient, environment-adaptive embodied agent generation. Inspired by recent co-optimization approaches (Yuan et al., 2021), BodyGen directly use auto-regressive transformers to generate an agent's morphology before executing environment interactions. Given an initial design (a.k.a a prompt), BodyGen can output optimal morphologies and corresponding controllers at the same time.

In contrast to previous methods, BodyGen utilizes a joint-level self-attention mechanism to achieve direct message communication using transformers. We further propose a topology-aware positional encoding for effective, lightweight morphology representation. Additionally, we addresses the inherent reward imbalance between morphology design (zero-reward-guided) and control (rich-reward-guided) phases: our enhanced temporal credit assignment mechanism dynamically balances reward signals across both stages, enabling coordinated optimization. The parameters of the whole BodyGen model is less than 2M, and a high-performance embodied agent can be generated using a single Nvidia GPU within 30 hours. To summarize, our contributions are as follows:

- We propose BodyGen, an end-to-end reinforcement learning framework for efficient embodiment co-design.
- We design a Morphology Self-Attention architecture (MoSAT) to provide joint-to-joint message transition, featuring our proposed Topological Position Encoding (TopoPE) for efficient morphology representation.
- We propose a temporal credit assignment mechanism that ensures balanced reward signals in the morphology design and control phases, thus facilitating co-design learning.

Comprehensive experiments across various tasks demonstrate BodyGen's advantages against previous methods in terms of both convergence speed and performance. BodyGen achieves an average performance improvement of **60.03%** against the state-of-the-art baselines.

## 2 RELATED WORK

**Universal Morphology Control** Embodiment co-design requires controlling robots with changeable morphologies and adapting to their incompatible action and state spaces. Universal Morphology Control (UMC), which employs a shared network to control each actuator separately, presents a promising solution to this problem. To better perceive the topological structures of various morphologies, some methods (Pathak et al., 2019; Wang et al., 2018a; Huang et al., 2020) employed Graph Neural Networks (GNNs) to enable communication between neighboring actuators. Recent works

also use Transformers (Vaswani et al., 2017) to overcome the limitations of multi-hop information aggregation brought by GNNs (Kurin et al., 2020; Hong et al., 2021; Gupta et al., 2021a; Dong et al., 2022). Despite these advancements, several challenges remain. Many existing methods are limited to parametric variations of a limited number (*e.g.* 2-3) of predefined morphologies, whereas a comprehensive morphology-agnostic design space remains largely unexplored. Furthermore, most previous works do not fully leverage morphology information or only consider its simple form and even the usefulness of such information remains controversial (Kurin et al., 2020; Hong et al., 2021; Gupta et al., 2021a; Xiong et al., 2023). In this work, we prove that morphology information plays a crucial role, and the *correctness* of morphology representation significantly influences performance. To this end, we introduce a simple yet effective positional encoding technique, TopoPE, which facilitates message localization within the body and enhances knowledge sharing among similar morphologies. By representing the 2D topological structure with position embeddings, we explore the potential of autoregressive transformers for robotic design generation.

**Embodiment Co-design**   As for embodied artificial agents, control policy (Lillicrap et al., 2015; Schulman et al., 2015a; Haarnoja et al., 2018; Schulman et al., 2017; Lowrey et al., 2018) has been well studied in the reinforcement learning and robotics community, while another critical component, the physical form of the embodiment, is currently attracting more and more attention (Kriegman et al., 2020; Bhatia et al., 2021; Xu et al., 2021; Huang et al., 2024). Embodiment co-design aims to optimize a robot's morphology and control simultaneously and is considered a promising way to stimulate the embodied intelligence embedded in morphology. Previous methods (Sims, 1994; Wang et al., 2018b; Gupta et al., 2021b) typically utilize evolutionary search (ES) to learn directly within the vast design space, which unavoidably brings inefficient sampling and expensive computation. A line of works (Wang et al., 2018b; Gupta et al., 2021a) introduces more human morphology priors, such as symmetry, to reduce the search space. Yuan et al. (2021) proposes jointly optimizing a robot's morphology and control policy via reinforcement learning. This paper focuses on the RL-based approach for joint optimization for both morphology and control. We aim to establish a comprehensive framework for embodiment co-design, systematically addressing key obstacles against efficiency during training.

## 3 PRELIMINARIES

**Morphology Representation.** The morphology of an agent can be formally defined as an undirected graph $\mathcal{G} = (V, E, A^v, A^e)$, where each node $v \in V$ represents a limb of the robot, and each edge $e = (v_i, v_j) \in E$ represents a joint connecting two limbs. $A^v$ and $A^e$ are two mapping functions that map the limb node $v$ to its physical attributes $A^v : V \to \Lambda^v$, and map the edge $e = (v_i, v_j)$ to its joint attributes $A^e : E \to \Lambda^e$, respectively. Here $\Lambda^v = \{\Lambda^{v_i}\}$ is the limb attribute space, consisting of attributes $\Lambda^{v_i}$ like limb lengths, sizes and materials, and $\Lambda^e = \{\Lambda^{e_i}\}$ is the joint attribute space consisting of attributes $\Lambda^{e_i}$ like rotation ranges and maximum motor torques. Consequently, the design space $D$ is defined on all valid robot morphologies $\mathcal{G} \in \mathcal{D}$.

**Co-Design Optimization.** The fitness $F$ of an agent represents its performance in a specific environment and is typically evaluated by rewards. In traditional control problems with a fixed morphology $\mathcal{G}_0$, we aim to optimize its control policy $\pi$ towards the optimal $\pi^* = \arg\max_\pi F(\pi, \mathcal{G}_0)$ for maximum fitness. In co-design problems, we not only optimize the control policy but also the morphology design simultaneously. This co-design process is formulated as a bi-level optimization problem:

$$\mathcal{G}^* = \arg\max_{\mathcal{G}} F(\pi^*_{\mathcal{G}}, \mathcal{G})$$
$$\text{s.t. } \pi^*_{\mathcal{G}} = \arg\max_{\pi_{\mathcal{G}}} F(\pi_{\mathcal{G}}, \mathcal{G}), \tag{3.1}$$

where the inner loop defines the optimal control policy of a given morphology, and the outer loop defines the optimal morphology using its optimal policy. Previous works typically use evolutionary algorithms (Sims, 1994; Wang et al., 2018b; Gupta et al., 2021b) to solve this problem. In this work, BodyGen leverages an RL-based framework and jointly optimizes both loops:

$$\pi^*(\cdot|\mathcal{G}^*), \mathcal{G}^* = \arg\max_{\pi(\cdot|\mathcal{G}),\mathcal{G}} F(\pi(\cdot|\mathcal{G}), \mathcal{G}), \tag{3.2}$$

using the universal control policy $\pi(\cdot|\mathcal{G})$, to facilitate knowledge sharing among agents with similar morphologies.

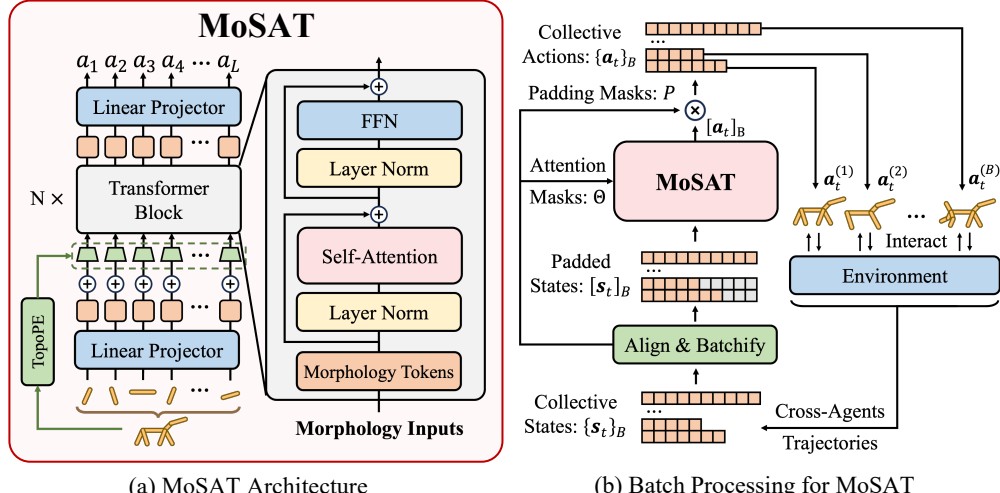

(a) MoSAT Architecture           (b) Batch Processing for MoSAT

Figure 3: **The Morphology Self-Attention (MoSAT) architecture**. (a) The sensor observations from different limbs are projected to hidden tokens for *centralized* processing with several MoSAT blocks and generate *separate* actions. (b) The MoSAT network processes different morphologies in a batch manner and learns a universal control policy $\pi(\cdot|\mathcal{G})$, thus improving training efficiency.

**Reinforcement Learning.** We define the problem formulation of Morphology-Conditioned Reinforcement Learning for embodiment co-design. We consider the augmented Markov Decision Process (MDP), which can be described by a 6-element tuple $\mathcal{M} = (\mathcal{S}, \mathcal{A}, \mathcal{T}, \mathcal{R}, \mathcal{D}, \Phi)$. $\Phi$ is a flag to distinguish design and control stages. $\mathcal{S}$ denotes the state space. $\mathcal{A}(\Phi)$ represents the action space, where $a \in \mathcal{A}(\Phi = \text{Design})$ changes the morphology of the agent, and $\mathcal{A}(\Phi = \text{Control})$ defines the action space for motion control. $\mathcal{T} : \mathcal{S} \times \mathcal{A}(\Phi) \times \mathcal{S} \to [0, 1]$ represents the environmental transitioning probability from one state $s_t$ to another $s_{t+1}$, given an action $a_t$. $\mathcal{R} : \mathcal{S} \times \mathcal{A} \times \mathcal{S} \to \mathbb{R}$ is the state-action reward, and the fitness function $F$ is defined as the episodic return $\sum_{t=1}^{T} r_t(s_t, a_t)$ based on rewards. As defined above, $\mathcal{D}$ represents the morphology design space, and our goal is to find some co-design policy $\pi : \mathcal{S} \times \mathcal{D} \to \mathcal{A}$ that can maximize the environmental fitness $F$.

## 4   METHOD

The co-design process consists of two sequential stages. (1) In the **Design Stage**, an agent begins with an initial morphology, $\mathcal{G}_0$, and iteratively refines through a series of morphology transforming actions via a design policy $\pi^D$, until it achieves the final design $\mathcal{G}_{done}$. In the subsequent (2) **Control Stage**, the agent interacts with the environment with its corresponding control policy $\pi^C$.

BodyGen addresses three key challenges that hinder co-design efficiency: (1) *Message Transmission Decay*, which occurs when multi-hop communication fails to effectively propagate information to distant limbs (Kurin et al., 2020). BodyGen leverages self-attention for both auto-regressively body building and centralized body control using transformers. (2) *Ineffective Morphology Representation* (Yuan et al., 2021; Hong et al., 2021). BodyGen employs a simple yet effective topology position encoding mechanism better to align similar morphologies for knowledge sharing between them. (3) *Unbalanced Reward Signals*. BodyGen utilizes a temporal credit assignment mechanism to ensure balanced reward signals between different co-design stages.

### 4.1   ATTENTION-BASED CO-DESIGN NETWORK

BodyGen divides the **Design Stage** into two sub-stages: *Topology Design Stage* and *Attribute Design Stage*, which transforms the topology $(V_0, E_0)$ and the corresponding attributes $(A_0^v, A_0^e)$ of the agent's morphology, respectively. Consequently, the design policy $\pi^D$ is also divided into two sub-policies $\pi^D = (\pi^{topo}, \pi^{attr})$ for according action control.

During the ***Topology Design Stage***, the agent can modify the topology through three basic actions: (1) `Addition`: add a new child limb $v_{new}$ to $v$, along with a new joint $e_{new} = (v, v_{new})$ connecting them. (2) `Deletion`: delete the limb $v$ and the joint to its parent $e = (v_p, v)$ if $v$ is a leaf node. (3) `NoChange`: take no changes for node $v$. The agent's policy $\pi^{topo}$ is conditioned on the current topology $(V, E)$ of timestep $t$, denoting as the product of action distributions $\pi_v^{topo}$ from all limbs [1]:

$$\boldsymbol{a}^{topo} \sim \pi^{topo}(\boldsymbol{a}^{topo}|\mathcal{G}) \triangleq \prod_{v \in V} \pi_v^{topo}(a_v^{topo}|p_v, \mathcal{G}) \tag{4.1}$$

where $p_v$ represents the topology position of node $v$.

In the ***Attribute Design Stage***, the agent further generates limb and joint attributes based on the given topology $\mathcal{G}_{done} = (V_{done}, E_{done})$. The agent's attribution policy $\pi^{attr}$ can be formulated as:

$$\boldsymbol{a}^{attr} \sim \pi^{attr}(\boldsymbol{a}^{attr}|\mathcal{G}) \triangleq \prod_{v \in V} \pi_v^{attr}(a_v^{attr}|p_v, \mathcal{G}) \tag{4.2}$$

Finally, in the **Control Stage**, the agent uses the morphology generated in the **Design Stage** to interact with the environment using the control policy $\pi^C$.

$$\boldsymbol{a}^{ctrl} \sim \pi^{ctrl}(\boldsymbol{a}^{ctrl}|\boldsymbol{s}, \mathcal{G}_{done}) \triangleq \prod_{v \in V} \pi_v^{ctrl}(a_v^{ctrl}|\boldsymbol{s}, p_v, \mathcal{G}_{done}), \tag{4.3}$$

where $\boldsymbol{s} = \{s_v\}$ denotes the sensor states of every limp, including forces, positions, velocities, *etc.* We use $a_v^{ctrl}$ to represent the torque of the joint connecting node $v$ with its parent $v_p$.

During the co-design process, we aim for the policy network to accommodate evolving morphologies in a way that offers two key advantages: (1) a single agent can maintain unified control even as the robot's body grows, preserving consistency across different designs, and (2) direct point-to-point communication between joints allows for richer information exchange, enabling more coordinated actions throughout the entire system. Inspired by the centralized signal processing of mammals in real-world nature, we propose the **Mo**rphology **S**elf-**A**tten**T**ion architecture (**MoSAT**) for efficient, centralized message processing. Figure 3 (a) provides an overview of MoSAT.

**Latent Projection.** We encode information from each limb's sensor to enhance network processing capabilities and map it into a latent space as *message tokens*. Specifically, limb sensor states $s_v$ are first processed through a parameter-shared linear mapping layer $\phi_h(\cdot)$:

$$\mathbf{m} = \phi_h(\mathbf{s}) + \mathbf{E}_{pos}(V, E) \quad \mathbf{s} \in \mathbb{R}^{L \times d}, \mathbf{m} \in \mathbb{R}^{L \times D} \tag{4.4}$$

where $d$ is the input state dimension and $D$ is the hidden dimension. We employ our proposed TopoPE for morphology representation, which will be further discussed later in Section 4.2. The position encodings $\mathbf{e}_v$ are added to message tokens $m_v$ to get position-embedded message tokens.

**Centralized Processing.** As illustrated in Figure 2, we aim for efficient message interaction. Body-Gen utilizes the scaled dot-product self-attention $\text{Attention}(\cdot)$ for point-to-point, centralized processing. Specifically, each message $\mathbf{m}$ use $q_{v_i}$ to query the key of another message $k_{v_j}$ weighting its value $v_{v_i}$:

$$\text{Attention}(\mathbf{m}) = \text{SoftMax}(\frac{QK^T}{\sqrt{d_k}})V \quad \text{where } Q = \mathbf{m}W_Q, K = \mathbf{m}W_K, V = \mathbf{m}W_V, \tag{4.5}$$

where $W_Q, W_K, W_V \in \mathbb{R}^{D \times D}$ are learnable matrices. For MoSAT block design, we adopt Pre-LN (Xiong et al., 2020) for layer normalization and add residual connections (He et al., 2016; Dosovitskiy et al., 2020).

**Forwarding.** In the end, we need to output actions for each actuator. We decode the attended messages using a linear projector $\phi_a(\cdot)$ to generate the action logits for each actuator:

$$\pi(\mathbf{a}|\mathbf{s}) = \begin{cases} \text{SoftMax}(\phi_a(\mathbf{m}_N)), & \text{Discrete Action Space} \\ \mathcal{N}(\mathbf{a}; \phi_a(\mathbf{m}_N), \Sigma), & \text{Continuous Action Space}. \end{cases} \tag{4.6}$$

---

[1]We use the limb-level action distribution, where each limb corresponds to its own action distribution, and the entire agent's action distribution is composed of all limbs' distributions. This effectively resolves the incompatibility of state and action spaces across the changeable topological morphologies.

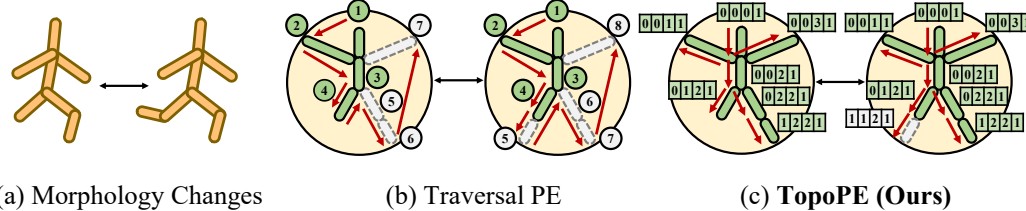

(a) Morphology Changes      (b) Traversal PE      (c) **TopoPE (Ours)**

Figure 4: The motivation of our proposed topology-aware position encoding TopoPE. (a) During the co-design procedure, the agent's morphology keeps changing. (b) A typical traversal-based PE in previous works resulted in inconsistency across mythologies. (c) TopoPE can better adapt to similar morphology structures using a reasonably alignable manner.

where $N$ is the stacked block number of the attention layer. The above process has equipped MoSAT with the capability to handle various morphologies. As shown in Figure 3 (b), to maximize training efficiency, we further offer MoSAT the ability to process multiple morphologies in a batch mode. We provide more implementation details in Appendix A.3.3.

### 4.2 TOPOLOGY-AWARE POSITION ENCODING FOR MORPHOLOGY REPRESENTATION

The vanilla attention operation treats each token equally, neglecting morphology information. However, it is crucial to inject positional information during embodiment co-design, for: (1) Similar information from different morphology positions has varying meanings and message source localization is significant; (2) Similar morphology structures may share similar local control policies, and positional information facilitates knowledge alignment and sharing across different agents. To better capture the differences between morphological structures and share structural knowledge among similar morphologies, we propose **Topology Position Encoding** (**TopoPE**), a topology-aware position encoding mechanism to handle the above two issues efficiently.

As demonstrated in Figure 4, for traversal-based limb indexing methods (Hong et al., 2021; Gupta et al., 2021a; Xiong et al., 2023) slight morphological changes can cause global indexing offsets. To mitigate the effect of offsets due to morphological changes, TopoPE uses a hash-map $\mathcal{H}(\cdot)$ for position encoding, which maps the path between the root limb $v_{root}$ and the current limb $v_i$ to a unique embedding $\mathbf{e}_{v_i}$:

$$\mathbf{e}_{v_i} = \mathcal{H}([v_i \mapsto v_{root}])$$
$$\text{where } [v_i \mapsto v_{root}] = [(v_i, p(v_i)), (p(v_i), p^2(v_i)), ..., (p^{l-1}(v_i), v_{root})], \tag{4.7}$$

where $p^n(v)$ is the $n$-th ancestor of $v$. Practically, if $v$ is the $k$-th child of its parent $p(v)$, the edge $(v, p(v))$ is denoted by the integer $k$, allowing the path index to be represented as a sequence of integers.

During the Topology Design Stage, BodyGen generates the robot's topology autoregressively (Figure 2). The topology created at each step is passed to MoSAT in the following step, where newly added limbs are automatically registered and assigned their Topology Position Embedding. Meanwhile, during the Attribute Design Stage and the Control Stage, this final topology remains fixed. Experiments demonstrate that TopoPE effectively adapts growing morphologies, facilitating knowledge alignment and sharing across agents, which leads to better performance.

### 4.3 CO-DESIGN OPTIMIZATION WITH TEMPORAL CREDIT ASSIGNMENT

To achieve efficient reward-driven co-design, Body-Gen leverages an actor-critic paradigm based on reinforcement learning, which trains a value function $V_\theta(s_t)$ and a policy function $\pi_\theta(a_t|s_t)$ and updates them using collected trajectories. We employ the Proximal Policy Optimization (PPO) (Schulman et al., 2017) to optimize the policy $\pi_\theta$ in the actor-critic framework. PPO uses the advantage function

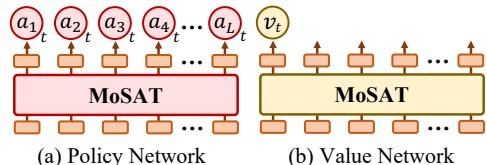

(a) Policy Network      (b) Value Network

Figure 5: BodyGen leverages an actor-critic paradigm for policy optimization.

$\hat{A}_t(a_t, s_t)$ to define how better an action $a_t$ is for current state $s_t$, and optimizes the following surrogate objective function as:

$$\mathcal{L}^{policy} = -\min\left\{\frac{\pi_\theta(a_t|s_t)}{\pi_{\theta_{old}}(a_t|s_t)}\hat{A}_t,\ \text{clip}\left(\frac{\pi_\theta(a_t|s_t)}{\pi_{\theta_{old}}(a_t|s_t)}, 1-\epsilon, 1+\epsilon\right)\hat{A}_t\right\}. \quad (4.8)$$

In the co-design process, vanilla PPO exhibits limited performance. Only the *Control Stage* directly receives environmental rewards, while theoretically, a body-modifying action in the Design Stage influences all future timesteps, whereas a motion-control action in the *Control Stage* has a diminishing impact over time. To address this, we decouple the MDPs for body and policy optimization, linking them through a modified Generalized Advantage Estimation (GAE) (Schulman et al., 2015b) for improved temporal credit assignment:

$$\hat{A}_t = \begin{cases} \delta_t + \gamma\lambda\hat{A}_{t+1} \cdot (1 - \mathbb{T}_t \vee \mathbb{C}_t), & \text{for Control Stage} \\ U_t - V_\theta(s_t), & \text{for Design Stage} \end{cases}$$

$$\text{where} \quad \delta_t = r_t + \gamma V_\theta(s_{t+1}) \cdot (1 - \mathbb{T}_t) - V_\theta(s_t)$$
$$U_t = r_t + U_{t+1} \cdot (1 - \mathbb{T}_t \vee \mathbb{C}_t), \quad (4.9)$$

where $\gamma$ is the discounting factor, $\lambda$ is the exponentially weighted for GAE and $\mathbb{T}_t, \mathbb{C}_t$ are two environment flags denoting environment termination and truncation, respectively. This decoupling enables us to apply distinct optimization algorithms to each stage, potentially improving overall performance. The value loss function $\mathcal{L}^{value}$ is defined as:

$$\mathcal{L}^{value} = \left(V_\theta(s_t) - \hat{R}_t\right)^2, \quad \text{where } \hat{R}_t = \text{sg}\left[V_\theta(s_t) + \hat{A}_t\right], \quad (4.10)$$

where $\text{sg}[\cdot]$ stands for the stop-gradient operator.

During the transition from the *Design Stage* to the *Control Stage*, we shift from a *GPT-style* (Radford et al., 2019) approach to a *BERT-style* (Devlin et al., 2018) framework. Specifically, the token output of each limb is used to generate the action policy for its corresponding actuator (Equation 4.6), as illustrated in Figure 5(a). Meanwhile, the token output of the root limb is used for value prediction of the entire body at timestep $t$ (Figure 5(b)). To prevent conflicts in gradient descent (Yu et al., 2020; Liu et al., 2021) arising from different credit assignment strategies, each stage in the co-design process is equipped with a separate value network.

## 5 EXPERIMENTAL EVALUATIONS

Our experiments aim to validate our primary hypothesis: that efficient message and reward delivery can effectively overcome bottlenecks in the co-design process, leading to embodied agents that can better adapt to the environment. Additional visualization results are presented in Appendix A.8. Visit our project website for more visualization results: https://genesisorigin.github.io.

**Environments.** We conduct a comprehensive evaluation of BodyGen with baselines in ten challenging co-design environments (CRAWLER, TERRAINCROSSER, CHEETAH, SWIMMER, GLIDER-REGULAR, GLIDER-MEDIUM, GLIDER-HARD, WALKER-REGULAR, WALKER-MEDIUM and WALKER-HARD) on MuJoCo (Todorov et al., 2012). These environments encompass diverse physical world types (2D, 3D), environment tasks, search space complexities, ground terrains, and initial designs to provide a multilevel evaluation. See Appendix A.1 for detailed descriptions.

### 5.1 COMPARISON WITH BASELINES

We compare BodyGen with the following baselines to highlight BodyGen's performance: 1) Evolution Based Algorithms: **NGE** (Wang et al., 2018b) maintains a population of agents with different morphologies for random mutation and only preserves top-performing agents' children for further optimization. 2) RL Based Algorithms: **Transform2Act** (Yuan et al., 2021) propose to optimize a robot's morphology and control concurrently through reinforcement learning and achieve co-optimization. It utilizes graph neural networks (GNNs) and joint-specific MLPs (JSMLP) to foster knowledge sharing and specification. 3) Universal Control Algorithms: **UMC-Message** (Wang et al., 2018a; Huang et al., 2020) leverages a localized message transition mechanism for information

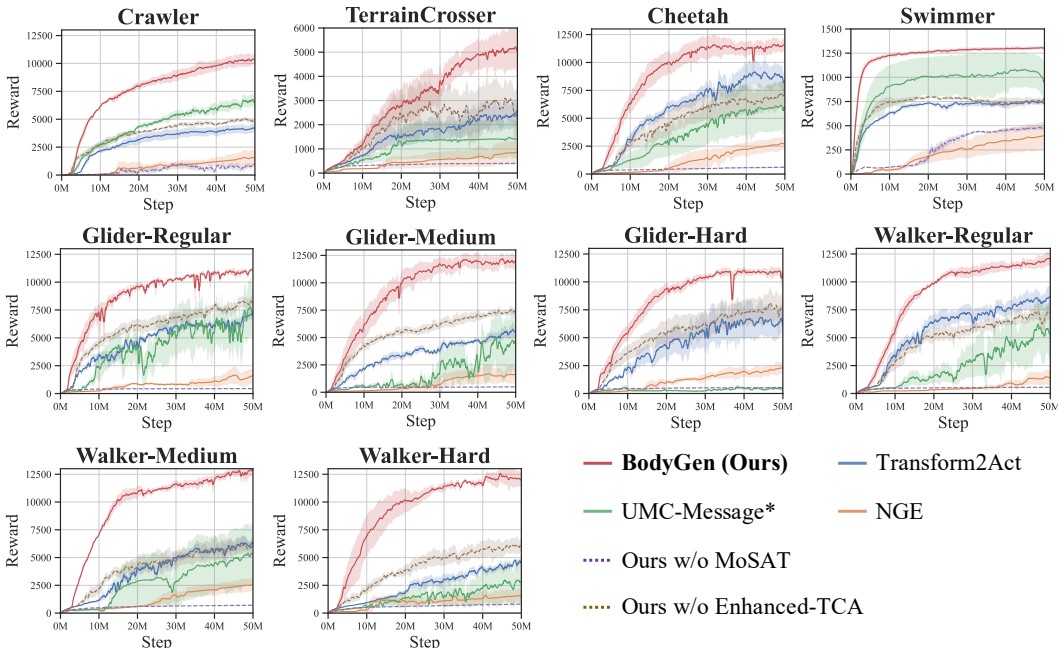

Figure 6: Performance of **BodyGen**, **NGE**, **Transform2Act**, **UMC-Message\***, **BodyGen w/o MoSAT**, and **BodyGen w/o Enhanced-TCA** on ten co-design environments, with error regions to indicate Standard Error over four random seeds.

exchange within the body, which is a typical method for universal morphology control. To make it suitable for embodiment co-design, we equip it with a policy network and our enhanced temporal credit assignment using reinforcement learning and denote it as UMC-Message*. The implementation details and full hyper-parameter of BodyGen and all baselines are provided in Appendix A.3 and Appendix A.4.

As shown in Figure 6, BodyGen achieves the highest task performance in all ten environments, with faster convergence speeds than baselines. Unlike the Universal Morphology Control (UMC) task, which focuses on limited specific morphologies (Wang et al., 2018a; Huang et al., 2020), embodiment co-design deals with various changeable, morphology-agnostic robots. Consequently, UMC-Message* fails to converge within a limited time for complex tasks such as GLIDER-HARD, and WALKER-HARD, due to its insufficient knowledge alignment mechanism for complicated, changable morphologies (*e.g.* JSMLP in Transform2Act and TopoPE in BodyGen).

Compared to evolutionary algorithms like NGE, we also find that RL-based methods demonstrate significant performance advantage due to a great sampling efficiency improvement within the same number of environmental interactions, supported by Yuan et al. (2021). By overcoming the bottlenecks in co-design, our approach goes even further: it achieves an average **60.03%** performance improvement over the strongest baseline in all the ten tasks.

## 5.2 ABLATION STUDIES

As mentioned in Section 4, our approach addresses inefficiencies in message and reward delivery, which includes the intra-agent level, inter-agent level, and agent-environment level. To better support our hypothesis and understand the importance of our key corresponding components (MoSAT, TopoPE, Enhanced-TCA), we designed four variants of our approach:

(i) *Ours w/o MoSAT*, which removes the MoSAT structure to remove our attention-based centralized information processing across different limbs; (ii) *Ours w/o Enhanced-TCA*, which removes our temporal credit assignment mechanism and employs original PPO for optimization; (iii) *Ours w/o TopoPE*, which removes TopoPE from our methods. For a more comprehensive comparison, we also introduced another position encoding method from recent UMC methods, as: (iv) *Ours w/*

Table 1: Comparison of different position encoding choices for morphology representation. The reported values are Mean $\pm$ Standard Error over four random seeds.

| Methods | CRAWLER | TERRAINCROSSER | CHEETAH | SWIMMER | GLIDER-REGULAR |
|---|---|---|---|---|---|
| **TopoPE (ours)** | **10381.96 $\pm$ 353.97** | **5056.01 $\pm$ 703.57** | **11611.52 $\pm$ 522.86** | 1305.17 $\pm$ 15.25 | **11082.29 $\pm$ 99.21** |
| w/ Traversal PE | 8582.24 $\pm$ 987.44 | 4339.60 $\pm$ 260.60 | 10581.62 $\pm$ 846.69 | 1292.05 $\pm$ 16.71 | 9801.31 $\pm$ 748.13 |
| w/o TopoPE | 7490.83 $\pm$ 267.70 | 1122.29 $\pm$ 659.38 | 7451.37 $\pm$ 2275.37 | **1371.20 $\pm$ 30.74** | 10137.83 $\pm$ 713.60 |

| Methods | GLIDER-MEDIUM | GLIDER-HARD | WALKER-REGULAR | WALKER-MEDIUM | WALKER-HARD |
|---|---|---|---|---|---|
| **TopoPE (ours)** | **11996.82 $\pm$ 595.51** | **10798.06 $\pm$ 298.39** | **12062.49 $\pm$ 513.07** | **12962.08 $\pm$ 537.34** | **11982.07 $\pm$ 520.78** |
| w/ Traversal PE | 10758.70 $\pm$ 401.90 | 9106.77 $\pm$ 679.59 | 10389.40 $\pm$ 1080.94 | 10972.13 $\pm$ 584.04 | 11255.89 $\pm$ 121.04 |
| w/o TopoPE | 4099.99 $\pm$ 2057.92 | 109.48 $\pm$ 10.03 | 10149.67 $\pm$ 255.99 | 6730.01 $\pm$ 705.06 | 6529.87 $\pm$ 1863.59 |

*Traversal PE*, where TopoPE is replaced with a traversal-based position embedding (Hong et al., 2021; Gupta et al., 2021a). Figure 6 presents the ablation studies for TopoPE and Enhanced-TCA, while Table 1 highlights the differences for different positional embedding choices. Additional detailed experimental results are available in the Appendix (Table 11, Table 12).

**(1) Intra-agent level**: The MoSAT module provides centralized information processing. Removing this module results in significant performance degradation. Transform2Act adds an MLP to each limb, enhancing local message processing and model performance, but it increases the model size to $19.64M$, which grows linearly with the complexity of the morphology. In contrast, BodyGen is more lightweight, with each model only with $1.43M$ parameters. We provide model parameters of BodyGen and baselines in Table 2.

**(2) Inter-agent level**: TopoPE facilitates morphological knowledge sharing among agents, aiding in adjusting knowledge for similar morphologies and reducing redundant learning costs. Compared to "Traversal PE" and "w/o TopoPE", TopoPE enhances agent performance and stabilizes learning.

**(3) Agent-environment level**: Our proposed temporal credit assignment ensures that an agent receives balanced reward signals during both morphology design and control phases, markedly improving final performance across all the environments for embodiment.

Table 2: Model parameters of BodyGen and baselines. *Note*: For NGE, the total number of models required is calculated as $20 + 20 \times 0.15 \times 125 = 395$ (population_size + population_size $\times$ elimination_rate $\times$ generations). The total parameters are derived with population_size only.

| **Models** | **Agent Parameters** | **Population Size** | **Total Parameters** |
|---|---|---|---|
| BodyGen (Ours) | 1.43 M | 1 | 1.43 M |
| Transform2Act | 19.64 M | 1 | 19.64 M |
| UMC-Message* | 0.27 M | 1 | 0.27 M |
| NGE | 0.27 M | 20 | 5.4 M |

# 6 CONCLUSIONS AND LIMITATIONS

This work proposes BodyGen, an end-to-end reinforcement learning framework for efficient embodiment co-design. Our approach delivers efficient messages and rewards through zero-decay message processing, effective morphological knowledge sharing, and balanced temporal credit assignment. Experiments demonstrate that BodyGen surpasses previous convergence speed and final performance methods while being efficient, lightweight, and scalable.

**Limitations and Future Work.** We acknowledge at least two limitations. Firstly, our approach remains focused on simulation environments, and further efforts are needed to transfer learned strategies to real physical systems. Secondly, our reward-driven reinforcement learning method focuses on improving control effects. Yet, it cannot simulate the rich perception and execution capabilities of real biological intelligent systems. In future research, we expect embodied intelligence to evolve perception and execution components akin to biological evolutionary principles, realizing more efficient tasks for embodied intelligence.

ACKNOWLEDGEMENT

This work was supported in part by the Natural Science Foundation of China under Grant No. 62222606 and the Ant Group Security and Risk Management Fund.

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

# A   APPENDIX

The supplementary material provides additional results, discussions, and implementation details.

Our code is available in our supplementary material for reproduction and further study. Visit our website for videos and more additional visualizations.

## A.1   ENVIRONMENT AND TASK DETAILS

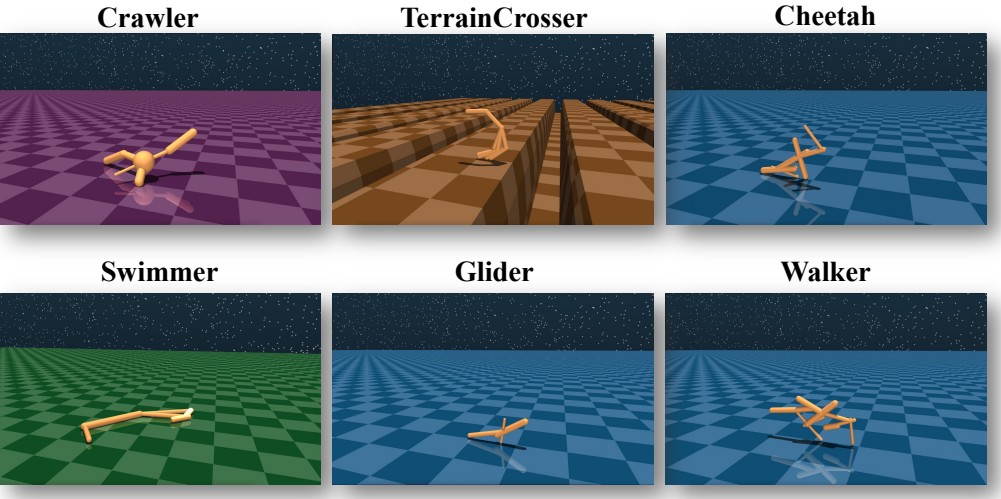

Figure 7: Randomly generated agents in six different environments for visualization. **Purple ground** indicates agents in a 3D physical world, **Green ground** represents agents in the xy-plane physical world, **Blue ground** denotes agents in the xz-plane physical world, and **Brown ground** denotes a physical world with variable terrain.

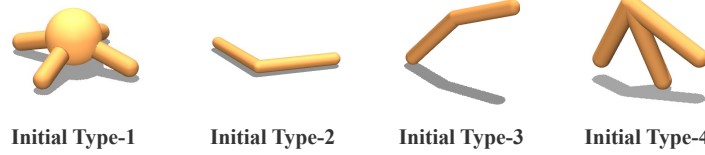

Figure 8: Visualization of four initial designs in the environments. *Type-1* consists of a structure with four limbs. *Type-2* and *Type-3* each includes two limbs connected by a joint, located in the xy-plane and xz-plane respectively. *Type-4* comprises three limbs connected by two joints. Note that BodyGen can support almost *arbitrary* initial designs and is not limited to specified types.

This section provides additional descriptions of the environments and tasks used in our experiments. Figure 7 displays randomly generated agents in six different environments. The first four environments: CRAWLER, TERRAINCROSSER, CHEETAH, and SWIMMER are derived from previous work (Yuan et al., 2021) to ensure a fair comparison. We have also introduced two additional environments, GLIDER and WALKER, to broaden the testing scope and provide a more comprehensive algorithm evaluation.

Each agent consists of multiple limbs connected by joints, each equipped with a motor for controlling movement. Sensors within the limbs monitor positional coordinates, velocity, and angular velocity. Each limb's attributes include limb length and limb size. Each joint's attributes cover rotation range and maximum motor torque. Each episode starts with a simple initial design, as demonstrated in Figure 8. The agent evolves to its final morphology through a series of topological and attribute modifications. Meanwhile, the control policy is required to optimize concurrently.

**Crawler** The agent inhabits a 3D environment with flat ground at $z = 0$. The initial design is the *Type-1* in Figure 8. Each limb can have up to two child limbs, except for the root limb. The height and 3D world velocity of the root limb are also included in the environment state. The reward function is defined as:

$$r_t = \frac{|p_{t+1}^x - p_t^x|}{\Delta t} - w \cdot \frac{1}{J} \sum_{u \in V_t} \|a_u^e\|^2 \tag{A.1}$$

where $w = 0.0001$ is the weighting factor for the control penalty term, $J$ is the total number of limbs, and $\Delta t = 0.04$.

**TerrainCrosser** The agent evolves in a terrain-variable environment, where the terrain features varying height differences. The maximum height difference of the terrain is $z_{max} = 0.5$. The agent must navigate these gaps to move forward. The initial design is the *Type-3* in Figure 8. The terrain is generated from a single-channel image, with different values representing different height rates. Each limb of the agent can have up to three child limbs. For the root limb, its height, 2D world velocity, and a variable encoding the terrain information are included in the environment state. The reward function is defined as:

$$r_t = \frac{|p_{t+1}^x - p_t^x|}{\Delta t}, \tag{A.2}$$

where $\Delta t = 0.008$, and the episode is terminated when the root limb height is below 1.0.

**Cheetah** The agent in this environment evolves with flat ground at $z = 0$. The initial design is the *Type-3* in Figure 8. Each limb of the agent can have up to three child limbs. The height and 2D world velocity of the root limb are added to the environment state. The reward function is defined in Equation (A.2). The episode is terminated when the root height is below 0.7.

**Swimmer** The swimmer is designed to cover snake-like creatures in the water. The agent evolves in water with a $vis = 0.1$ viscosity for water simulation. The initial design is the *Type-2* in Figure 8. Each limb supports up to three child joints. The root limb's 2D world velocity is incorporated into the environment state. The reward function is the same as TerrainCrosser in Equation (A.2).

**Glider** The agent in this environment evolves on flat ground. The initial design is the *Type-4*, as shown in Figure 8. In Glider, the agent's search depth is limited to three times that of the initial design, encouraging full exploration of a relatively shallow search space. We also provide three different task levels: regular, medium, and hard, where each limb of the agent can have up to one, two, or three child limbs. The reward function is defined in Equation (A.2).

**Walker** The agent evolves on flat ground. The starting design is *Type-4* in Figure 8. The search depth for the agent is capped at four times the initial design to promote thorough exploration within a comparatively shallow search space. Similarly, three levels of task difficulty are offered, which have the same meaning as described in Glider. The reward function is specified in Equation (A.2).

Note that the reward functions are kept simple and consistent in all environments. Unlike common practices in OpenAI-Gym (Brockman et al., 2016), we do not provide any additional reward priors (*e.g.*alive bonus) to facilitate learning, which presents higher requirements to the algorithm robustness.

## A.2 MOTIVATIONS

This section will detail the motivations behind designing MoSAT and TopoPE for embodiment co-design, aiming to provide further insights.

### A.2.1 CENTRALIZED MESSAGE PROCESSING AND MOSAT

As demonstrated in Figure 9, GNN-like neural systems are commonly found in simple organisms such as planarians, where sensory information is connected through neural networks for distributed and localized processing. In contrast, advanced creatures such as humans utilize a centralized signal processing approach, where signals from various body parts are centrally processed in the brain, leveraging scalability advantages similar to the self-attention mechanism within transformers.

Figure 10 further illustrates the different message delivery mechanisms between GNN and Transformer. GNN uses aggregation and broadcasting for message transmission, resulting in progressive

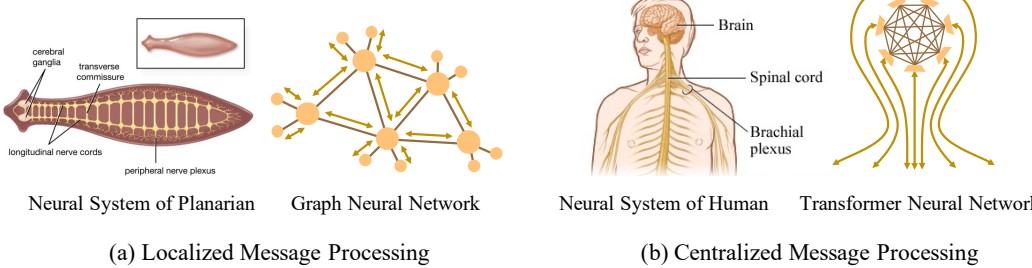

(a) Localized Message Processing      (b) Centralized Message Processing

Figure 9: Comparative overview of natural and artificial neural processing systems. (a) Localized message processing in planarians and GNNs. (b) Centralized message processing in human brains and Transformers. Relevant images are sourced from Encyclopaedia Britannica (2024); Brain for AI Fandom (2024).

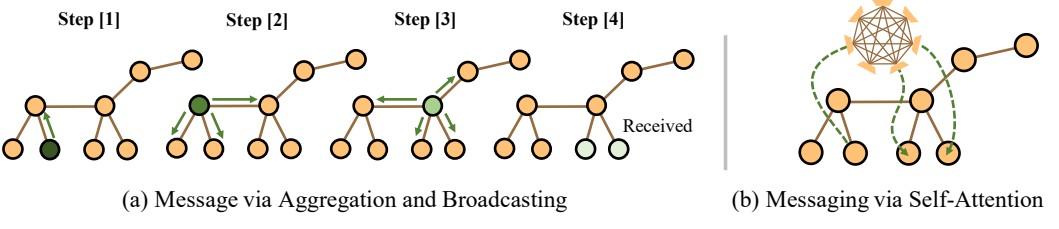

(a) Message via Aggregation and Broadcasting      (b) Messaging via Self-Attention

Figure 10: Comparison of different message delivery mechanisms between GNN and Transformer.

information reduction. As demonstrated in Figure 10 (a), the dog-like robot needs to adjust its posture throughout its motion. The GNN's localized message processing approach requires signals from distant locations to propagate multiple times before reaching the target actuator. In contrast, Transformers can provide faster message transfer and interaction, by employing self-attention to facilitate direct point-to-point and point-to-multipoint message delivery. Inspired by this, we propose MoSAT in Section 4.1. MoSAT first maps sensor information to the latent space and leverage self-attention for signal interactions for centralized decision-making.

Meanwhile, in GNNs, the message propagation mechanism allows for an implicit representation of morphology. However, while transformers leverage self-attention for direct message delivery, they do not offer an asymmetric information propagation mechanism to differentiate positions between different body parts.

### A.2.2 MORPHOLOGY POSITION EMBEDDING AND TOPOPE

Position encoding has proven effective in location representation within the natural language processing field (Vaswani et al., 2017; Shaw et al., 2018; Raffel et al., 2020; Wang et al., 2019).

Effectively representing the robot's morphology is crucial for co-designing morphology and control policies. In our work, we propose the **Topology Position Embedding (TopoPE)** to encode the morphology in a way compatible with Transformer-based architectures. TopoPE assigns a unique embedding to each limb based on its topological position within the robot's morphology tree. Specifically, the embedding index for a limb is derived from the path from the root node to the limb, capturing the structural relationships within the morphology.

In previous works (Trabucco et al., 2022; Gupta et al., 2021a), morphology encodings often rely on traversal sequences like depth-first search (DFS) or manual naming (Trabucco et al., 2022; Li et al., 2024) conventions based on a "full model" of the robot. When limbs are removed to generate variants, the names of the remaining limbs remain unchanged, facilitating consistent encoding.

However, in our setting, there is no predefined "full model," and the robot's morphology is dynamically generated during co-design. Manually naming limbs is impractical in this context. Our TopoPE addresses this challenge using a topology indexing mechanism, which uses the path to the root as

the embedding index. This method naturally extends to dynamically changing morphologies and ensures that similar substructures share similar embeddings, promoting generalization across different morphologies.

Moreover, unlike learnable position embeddings that are specific to particular morphologies, our approach can be extended using non-learnable embeddings, such as sinusoidal embeddings (Vaswani et al., 2017), which offer better extrapolation to unseen morphologies and eliminate the need for training the embeddings.

To demonstrate the effectiveness of TopoPE, we conducted ablation studies comparing models with and without TopoPE. As shown in Table 1 and Figure 11, incorporating TopoPE significantly improves performance across various tasks. This indicates that TopoPE provides a more informative and stable encoding of the morphology, facilitating better learning of control policies.

In contrast to other morphology-aware positional encodings, our TopoPE is specifically designed to handle dynamic and diverse morphologies without relying on a fixed full model or manual limb naming. Additionally, our approach aligns well with the Transformer architecture, allowing standard attention mechanisms to capture interactions between different limbs based on their topological relationships.

## A.3 Implementation Details

### A.3.1 Training Details

In line with standard reinforcement learning practices, we employed distributed trajectory sampling across multiple CPU threads to accelerate training. Each model is trained using four random seeds on a system equipped with 112 Intel® Xeon® Platinum 8280 cores and six Nvidia RTX 3090 GPUs. Our main code framework is based on Python 3.9.18 and PyTorch 2.0.1. For all the environments used in our work, it takes approximately only 30 hours to train a model with 20 CPU cores and a single NVIDIA RTX 3090 GPU on our server.

### A.3.2 Hyperparameters

For BodyGen, we ran a grid search over MoSAT layer normalization $\in$ $\{$w/o-LN, Pre-LN, Post-LN$\}$, Policy network learning rate $\in$ $\{5e-5, 1e-4, 3e-4\}$, Value network learning rate $\in \{1e-4, 3e-4\}$, and MoSAT hidden dimension $\in \{32, 64, 128, 256\}$. We did not search further for the environmental settings, optimizer configurations, PPO-related hyperparameters, or the training batch and minibatch sizes. Instead, we strictly maintained consistency with previous works (Wang et al., 2018b; Yuan et al., 2021; Kurin et al., 2020) to ensure a fair comparison. With further hyperparameter tuning, our algorithm could achieve higher performance levels. Table 3 displays the hyperparameters BodyGen adopted across all experiments.

For Transform2Act, we followed previous work (Yuan et al., 2021) and its official released code repository [2], and used $\mathrm{GraphConv}$ as the GNN layer type, policy GNN size $(64, 64, 64)$, policy learning rate $5e-5$, value GNN size $(64, 64, 64)$, value learning rate $3e-4$, JSMLP activation function $\mathrm{Tanh}$, JSMLP size $(128, 128, 128)$ for the policy, MLP size $(512, 256)$ for the value function, which were the best values they picked using grid searches.

To make UMC-Message suitable for embodiment co-design, we equip them with a policy network and employ our temporal credit assignment via reinforcement learning. The network parameters and training settings are consistent with those used in BodyGen and Transform2Act to ensure a fair comparison. It adopted GNN layer type of $\mathrm{GraphConv}$, policy GNN size $(64, 64, 64)$, policy MLP size $(128, 128)$, policy learning rate $5e-5$, value GNN size $(64, 64, 64)$, value MLP size $(512, 256)$, value learning rate $3e-4$. We followed previous work (Huang et al., 2020) and also referred to the publicly released code [3] for implementation.

For NGE, we follow previous works (Wang et al., 2018b; Yuan et al., 2021) according to the public release code [4], and used a number of generations 125, agent population size 20, elimination rate 0.15,

---

[2] https://github.com/Khrylx/Transform2Act
[3] https://github.com/huangwl18/modular-rl
[4] https://github.com/WilsonWangTHU/neural_graph_evolution

GNN layer type $\mathrm{GraphConv}$, MLP activation $\mathrm{Tanh}$, policy GNN size $(64, 64, 64)$, policy MLP size $(128, 128)$, value GNN size $(64, 64, 64)$, value MLP size $(512, 256)$, policy learning rate $5e-5$, and value learning rate $3e-4$, which were the best searched values described by previous work.

Table 3: Hyperparameters of BodyGen adopted in all the experiments

| Hyperparameter | Value |
|---|---|
| Number of Topology Design $N^{topo}$ | 5 |
| Number of Attribute Design $N^{attr}$ | 1 |
| MoSAT Layer Normalization | Pre-LN |
| MoSAT Activation Function | SiLu |
| MoSAT FNN Scaling Ratio $r$ | 4 |
| MoSAT Block Number (Policy Network) | 3 |
| MoSAT Block Number (Value Network) | 3 |
| MoSAT Hidden Dimension (Policy Network) | 64 |
| MoSAT Hidden Dimension (Value Network) | 64 |
| Optimizer | Adam |
| Policy Learning Rate | 5e-5 |
| Value Learning Rate | 3e-4 |
| Clip Gradient Norm | 40.0 |
| PPO Clip $\epsilon$ | 0.2 |
| PPO Batch Size | 50000 |
| PPO Minibatch Size | 2048 |
| PPO Iterations Per Batch | 10 |
| Training Epochs | 1000 |
| Discount factor $\gamma$ | 0.995 |
| GAE $\lambda$ | 0.95 |

### A.3.3 THE BATCH MODE FOR MOSAT

To maximize training efficiency, we further offer MoSAT the ability to process multiple morphologies in a batch mode. For a batch of state inputs $\{s_t\}_B$, we first pad them to equal length $[s_t]_B \in \mathbb{R}^{B \times L_m \times d}$, where $L_m$ is the max limb number of morphologies within this batch, and generate a padding matrix $P \in \mathbb{R}^{B \times L_m}$, where $P_{ij} = 1$ for $j \leq L_i$ and $P_{ij} = 0$ for $j > L_i$. To keep the messaging logic exactly equivalent to the regular mode, we can eliminate the influence of padding by modifying the attention operation with an attention mask $\Theta \in \mathbb{R}^{B \times L_m \times L_m}$:

$$\mathrm{Attention}([\mathbf{m}_t]_B) = \mathrm{SoftMax}(\frac{QK^T}{\sqrt{d_k}} + \Theta)V, \tag{A.3}$$

where $\Theta_{ijk} = \log(P_{ik} + \epsilon)$. Finally, we remove the batch padding and re-allocate actions to joints of different agents via: $\{\boldsymbol{a}\}_B = [\boldsymbol{a}]_B \odot P$, where $\odot$ represents the bool-selection operation according to the padding matrix $P$.

### A.4 ALGORITHM DETAILS

Algorithm 1 illustrates the overall training process of BodyGen, which is based on PPO for efficient reinforcement learning. We highlight three key components: the interaction process, our temporal credit assignment based on GAE, and the main loop for iterative optimization.

---

**Algorithm 1:** Synchronous Learning Algorithm for BodyGen

---

**Input:** Replay Buffer $\mathbf{B}$, Batch $\mathbb{B}$, Optimizer $optimizer$

**Initialize :** Policy networks: $\pi_\theta : \{\pi_\theta^{topo}, \pi_\theta^{attr}, \pi_\theta^{ctrl}\}$; Value networks: $V_\theta : \{V_\theta^{topo}, V_\theta^{attr}, V_\theta^{ctrl}\}$

$\qquad\qquad$ $\mathbf{B} \leftarrow \varnothing, \mathbb{B} \leftarrow \varnothing$, Discount factor $\gamma$, GAE Exponential Weight $\lambda$

1 **Function** INTERACT(Policy: $\pi$, Replay Buffer: $\mathbf{B}$)**:**

2 $\quad$ **while** $\mathbf{B}$ *not reaching max buffer size* **do**

3 $\qquad$ $\mathcal{G}_0 \leftarrow$ initial design

4 $\qquad$ $\Phi \leftarrow topo$ $\hfill \triangleright$ Topology design stage

5 $\qquad$ **for** $t = 0, 1, ..., N^{topo} - 1$ **do**

6 $\qquad\quad$ $\boldsymbol{a}_t^{topo} \sim \pi^{topo}(\boldsymbol{a}_t^{topo}|\mathcal{G}_t)$ $\hfill \triangleright$ Sample topology actions from all limps

7 $\qquad\quad$ $\mathcal{G}_{t+1} \leftarrow$ apply $\boldsymbol{a}_t^{topo}$ to modify the topology $(V_t, E_t)$ of current design $\mathcal{G}_t$

8 $\qquad\quad$ $r_t = 0$ ; $\mathcal{S}_t = \mathcal{S}$; store $\{r_t, \varnothing, \boldsymbol{a}_t^{topo}, \mathcal{G}_t, \mathcal{S}_t, 0, 0\}$ into $\mathbf{B}$ $\hfill \triangleright$ Update Buffer $\mathbf{B}$ with transition

9 $\qquad$ **end**

10 $\qquad$ $\Phi \leftarrow attr$ $\hfill \triangleright$ Attribute design stage

11 $\qquad$ **for** $t = N^{topo}, ..., N^{topo} + N^{attr} - 1$ **do**

12 $\qquad\quad$ $\boldsymbol{a}_t^{attr} \sim \pi^{attr}(\boldsymbol{a}_t^{attr}|\mathcal{G}_t)$ $\hfill \triangleright$ Sample attribute actions from all limps

13 $\qquad\quad$ $\mathcal{G}_{t+1} \leftarrow$ apply $\boldsymbol{a}_t^{attr}$ to modify the attribute $(A_t^v, A_t^e)$ of current design $\mathcal{G}_t$

14 $\qquad\quad$ $r_t = 0$ ; $\mathcal{S}_t = \mathcal{S}$; store $\{r_t, \varnothing, \boldsymbol{a}_t^{attr}, \mathcal{G}_t, \mathcal{S}_t, 0, 0\}$ into $\mathbf{B}$ $\hfill \triangleright$ Update Buffer $\mathbf{B}$ with transition

15 $\qquad$ **end**

16 $\qquad$ $\Phi \leftarrow ctrl$ $\hfill \triangleright$ Control stage

17 $\qquad$ $\boldsymbol{s}_t \leftarrow$ Env.Reset(0) $\hfill \triangleright$ $\boldsymbol{s}_t = \{s_{v,t}\}$ denotes the sensor states from all limps

18 $\qquad$ **for** $t = N^{topo} + N^{attr}, ..., T - 1$ **do**

19 $\qquad\quad$ $\boldsymbol{a}_t^{ctrl} \sim \pi^{ctrl}(\boldsymbol{a}_t^{ctrl}|\boldsymbol{s}_t, \mathcal{G}_{done})$

20 $\qquad\quad$ $r_t, \boldsymbol{s}_{t+1}, \mathbb{T}_t, \mathbb{C}_t \leftarrow$ Env.Step($\boldsymbol{a}_t^{ctrl}$) $\hfill \triangleright$ $\mathbb{T}_t, \mathbb{C}_t$ denotes termination and trunction

21 $\qquad\quad$ $\mathcal{S}_t = \mathcal{S}$; store $\{r_t, \boldsymbol{s}_t, \boldsymbol{a}_t^{ctrl}, \mathcal{G}_t, \mathcal{S}_t, \mathbb{T}_t, \mathbb{C}_t\}$ into $\mathbf{B}$ $\hfill \triangleright$ Update Buffer $\mathbf{B}$ with transition

22 $\qquad$ **end**

23 $\quad$ **end**

24 **end**

25 **Function** ENHANCEDGAE(Value Function: $V_\theta$, Replay Buffer: $\mathbf{B}$)**:**

26 $\quad$ **for** $t = T - 1, ..., 0$ **do**

27 $\qquad$ $U_t = r_t + U_{t+1} \cdot (1 - \mathbb{T}_t \vee \mathbb{C}_t)$ $\hfill \triangleright$ Calculate return

28 $\qquad$ $\delta_t = r_t + \gamma V_\theta(s_{t+1}) \cdot (1 - \mathbb{T}_t) - V_\theta(s_t)$ $\hfill \triangleright$ Calculate the TD-error term

29 $\qquad$ **if** $\mathcal{S}_t = ctrl$ **then**

30 $\qquad\quad$ $\hat{A}_t = \delta_t + \gamma\lambda\hat{A}_{t+1} \cdot (1 - \mathbb{T}_t \vee \mathbb{C}_t)$ $\hfill \triangleright$ Calculate advantage for the control stage

31 $\qquad$ **else**

32 $\qquad\quad$ $\hat{A}_t = U_t - V_\theta(s_t)$ $\hfill \triangleright$ Calculate advantage for the design stage

33 $\qquad$ **end**

34 $\qquad$ $\hat{R}_t = V_\theta(s_t) + \hat{A}_t$ $\hfill \triangleright$ Calculate the target value

35 $\qquad$ store $\{\hat{A}_t, \hat{R}_t\}$ into $\mathbf{B}$ $\hfill \triangleright$ Append $\hat{A}_t$ and $\hat{R}_t$ to the corresponding transition item in $\mathbf{B}$.

36 $\quad$ **end**

37 **end**

38 **Function** MAIN()**:**

39 $\quad$ **while** *not reaching max iterations* **do**

40 $\qquad$ $Th_i \leftarrow$ Thread(INTERACT, $\pi_\theta$, $\mathbf{B}$) $\hfill \triangleright$ We use multiple CPU threads for sampling

41 $\qquad$ $Th_i.join()$ $\hfill \triangleright$ Gather trajectories collected from threads

42 $\qquad$ ENHANCEDGAE($V_\theta$, $\mathbf{B}$) $\hfill \triangleright$ Perform temporal credit assignment for co-design

43 $\qquad$ **while** *not reaching max epochs* **do**

44 $\qquad\quad$ Update $\mathbb{B} \leftarrow \mathbf{B}$ $\hfill \triangleright$ Sample a random batch $\mathbb{B}$ from Buffer $\mathbf{B}$

45 $\qquad\quad$ Calculate PPO loss $\mathcal{L}_{ppo} = \mathcal{L}^{policy} + \mathcal{L}^{value}$ $\hfill \triangleright$ According to Equation (4.8) and (4.10)

46 $\qquad\quad$ $optimizer \leftarrow$ Gradient from $\mathcal{L}_{ppo}$ $\hfill \triangleright$ Gradient descent to update $\pi_\theta$ and $V_\theta$

47 $\qquad$ **end**

48 $\quad$ **end**

49 **end**

---

## A.5 ADDITIONAL RESULTS

### A.5.1 QUANTITATIVE RESULTS

Table 4: Comparison of BodyGen, its ablation variants, and baseline methods.

| Methods | CRAWLER | TERRAINCROSSER | CHEETAH | SWIMMER | GLIDER-REGULAR |
|---|---|---|---|---|---|
| **BodyGen (Ours)** | **10381.96 ± 353.97** | **5056.01 ± 703.57** | **11611.52 ± 522.86** | **1305.17 ± 15.25** | **11082.29 ± 99.21** |
| - w/o MoSAT | 818.92 ± 57.78 | 407.30 ± 4.50 | 662.88 ± 74.88 | 476.26 ± 19.95 | 447.72 ± 7.56 |
| - w/o Enhanced-TCA | 4994.44 ± 160.14 | 2668.66 ± 844.22 | 8158.74 ± 55.71 | 786.32 ± 19.39 | 8317.88 ± 498.26 |
| Transform2Act | 4185.63 ± 334.04 | 2393.84 ± 692.96 | 8405.70 ± 815.64 | 732.20 ± 22.61 | 6901.68 ± 374.42 |
| NGE | 1545.13 ± 626.54 | 881.71 ± 459.96 | 2740.79 ± 515.51 | 395.90 ± 173.85 | 1567.84 ± 756.74 |
| UMC-Message | 6492.90 ± 441.04 | 1411.51 ± 705.68 | 5785.40 ± 2110.77 | 961.20 ± 183.03 | 7354.34 ± 2145.22 |
| Methods | GLIDER-MEDIUM | GLIDER-HARD | WALKER-REGULAR | WALKER-MEDIUM | WALKER-HARD |
| **BodyGen (Ours)** | **11996.82 ± 595.51** | **10798.06 ± 298.39** | **12062.49 ± 513.07** | **12962.08 ± 537.34** | **11982.07 ± 520.78** |
| - w/o MoSAT | 489.75 ± 5.74 | 533.17 ± 14.20 | 555.33 ± 18.15 | 708.32 ± 12.72 | 827.33 ± 47.71 |
| - w/o Enhanced-TCA | 7454.55 ± 289.93 | 7592.03 ± 1023.70 | 7286.30 ± 735.55 | 6069.51 ± 652.96 | 6126.73 ± 572.85 |
| Transform2Act | 5573.44 ± 519.22 | 6120.37 ± 1380.74 | 8685.47 ± 1008.88 | 6287.15 ± 426.99 | 4645.31 ± 294.81 |
| NGE | 1649.60 ± 763.55 | 2339.90 ± 487.22 | 1402.85 ± 595.54 | 2600.39 ± 481.74 | 1575.87 ± 508.11 |
| UMC-Message | 4726.44 ± 2406.35 | 425.49 ± 141.02 | 5417.14 ± 2019.43 | 5347.70 ± 2397.85 | 2783.09 ± 1587.06 |

As demonstrated in Figure 6, we present the full training curves for BodyGen with baselines including Transform2Act, UMC-Message, NGE, and ablation variants of *ours w/o MoSAT* and *ours w/o Enhanced-TCA* across ten co-design environments. Each model was trained using four random seeds. For all baselines, we employed the best performance configurations reported by previous works, as is detailed in Section A.3. Table 4 further presents related metrics, with each cell showing the mean and standard deviation of episode rewards for the corresponding algorithm in each environment.

## A.6 ADDITIONAL ABLATION STUDIES ON TOPOPE AND ENHANCED-TCA

We provide additional ablation studies on our proposed TopoPE and Enhanced-TCA to provide more insights, demonstrated in Figure 11 and Figure 12.

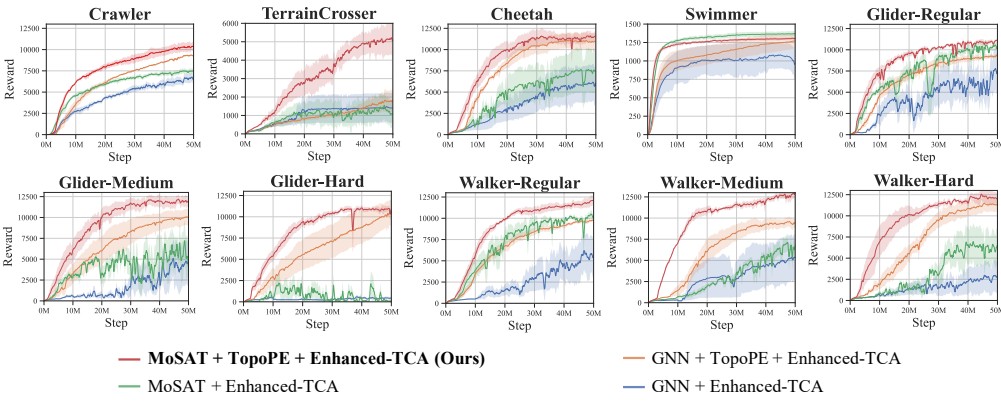

Figure 11: Extensive experiments on our proposed simple-yet-effective Topology Position Encoding (TopoPE) across different architectures of MoSAT and GNN, validating TopoPE as an efficient and general method for morphology representation. (1) MoSAT: w/o TopoPE → with TopoPE; (2) GNN: w/o TopoPE → with TopoPE; Both sets demonstrated the obvious performance improvements brought by TopoPE.

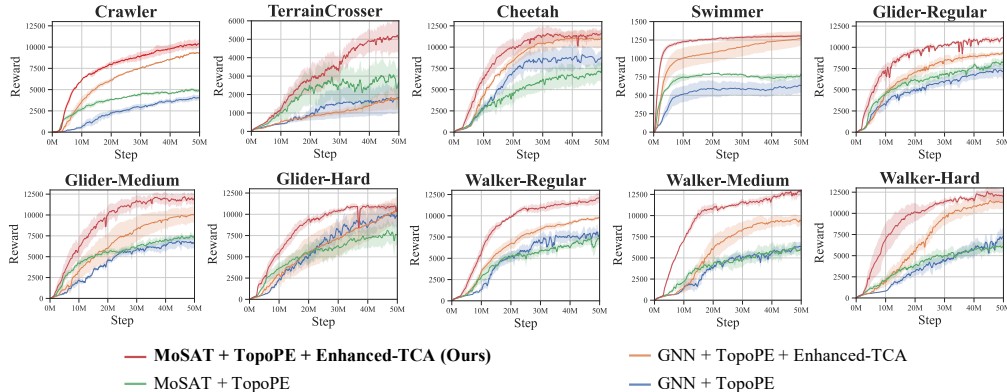

Figure 12: Extensive experiments on our proposed Temporal Credit Assignment Mechanism (Enhanced-TCA) across different architectures of MoSAT and GNN, validating Enhanced-TCA mechanism as an efficient method for enhancing bi-level optimization. (1) MoSAT: w/o Enhanced-TCA → with Enhanced-TCA; (2) GNN: w/o Enhanced-TCA → with Enhanced-TCA; Both sets demonstrated the obvious performance improvements brought by our Enhanced-TCA mechanism.

## A.7 COMPARISON OF BODYGEN'S DESIGN SPACE WITH UNIMAL

In addition to better position BodyGen, we also compare its design space and computational requirements to those of UNIMAL (Gupta et al., 2021b), a widely recognized framework for morphology design. BodyGen and UNIMAL (Gupta et al., 2021b) share similarities and differences in their approaches to morphology design, search space, and computational demands, providing insights into the trade-offs between these systems. We will compare them from several perspectives:

**Initial design.** The search space of UNIMAL is similar with the design space in our "crawler" environment. Both BodyGen (in the crawler environment) and UNIMAL adopt an ant-like structure with a single body and four limbs extending in perpendicular directions, as the initial design $\mathcal{G}_0$.

**Morphology actions.** UNIMAL offers three basic mutation operations: adding limbs, deleting limbs, and modifying limb parameters. BodyGen employs a similar set of actions but organizes them into two types: the topology design type, which includes adding limbs, deleting limbs, and passing, and the attribute design type, which focuses on modifying limb parameters. While these actions are conceptually aligned, BodyGen provides a more structured framework for exploring morphology changes.

**Search space.** UNIMAL allows for a maximum of 10 limbs, whereas the "crawler" environment used by BodyGen supports up to 29 limbs, offering a significantly larger space for morphological exploration. This difference highlights BodyGen's broader scope in accommodating complex designs.

## A.8  MORE VISUALIZATION RESULTS

In this section, we provide additional visualization results for embodied agents generated by BodyGen across ten co-design environments.

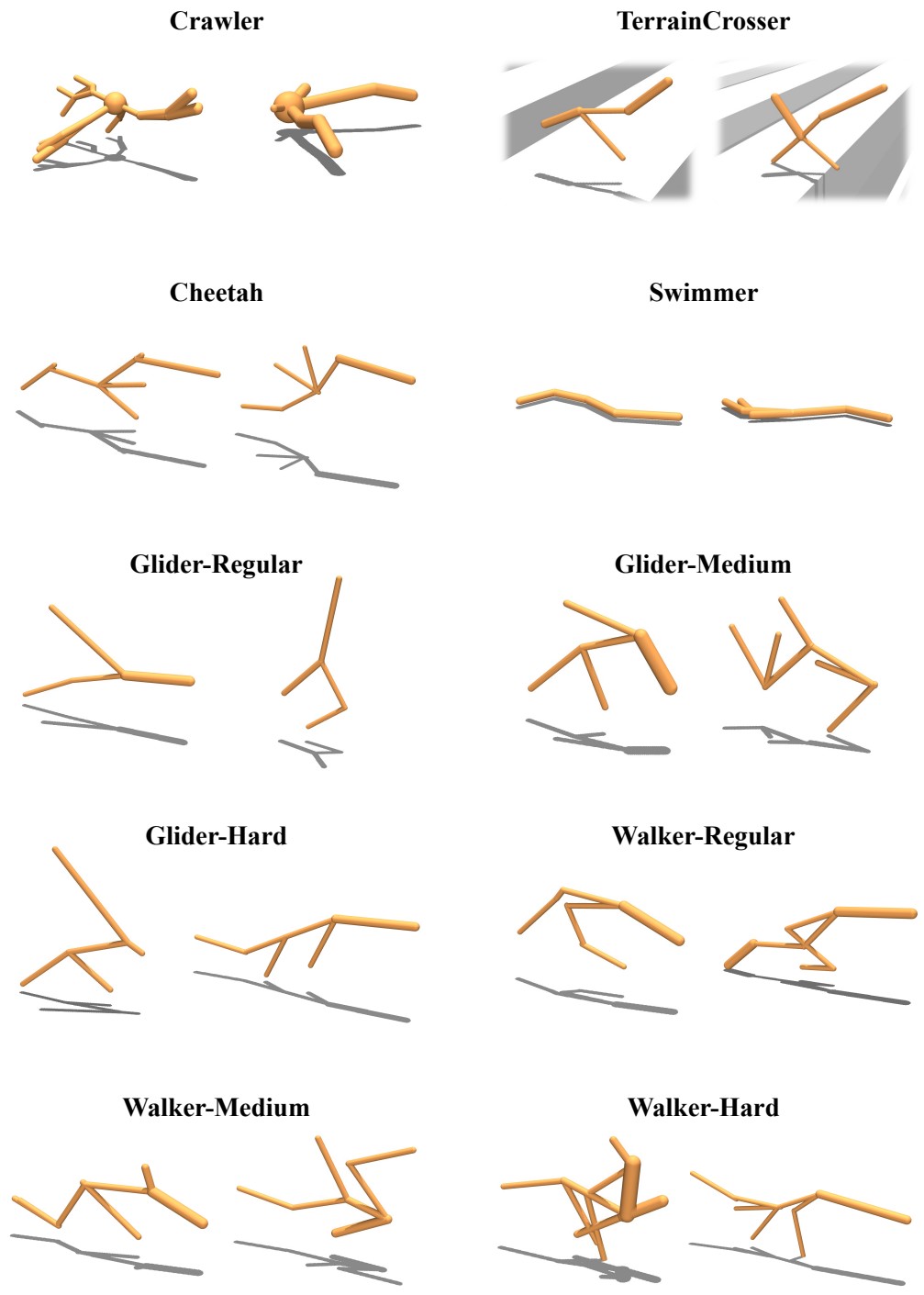

Figure 13: Visualization of embodied agents generated by BodyGen on different environments.

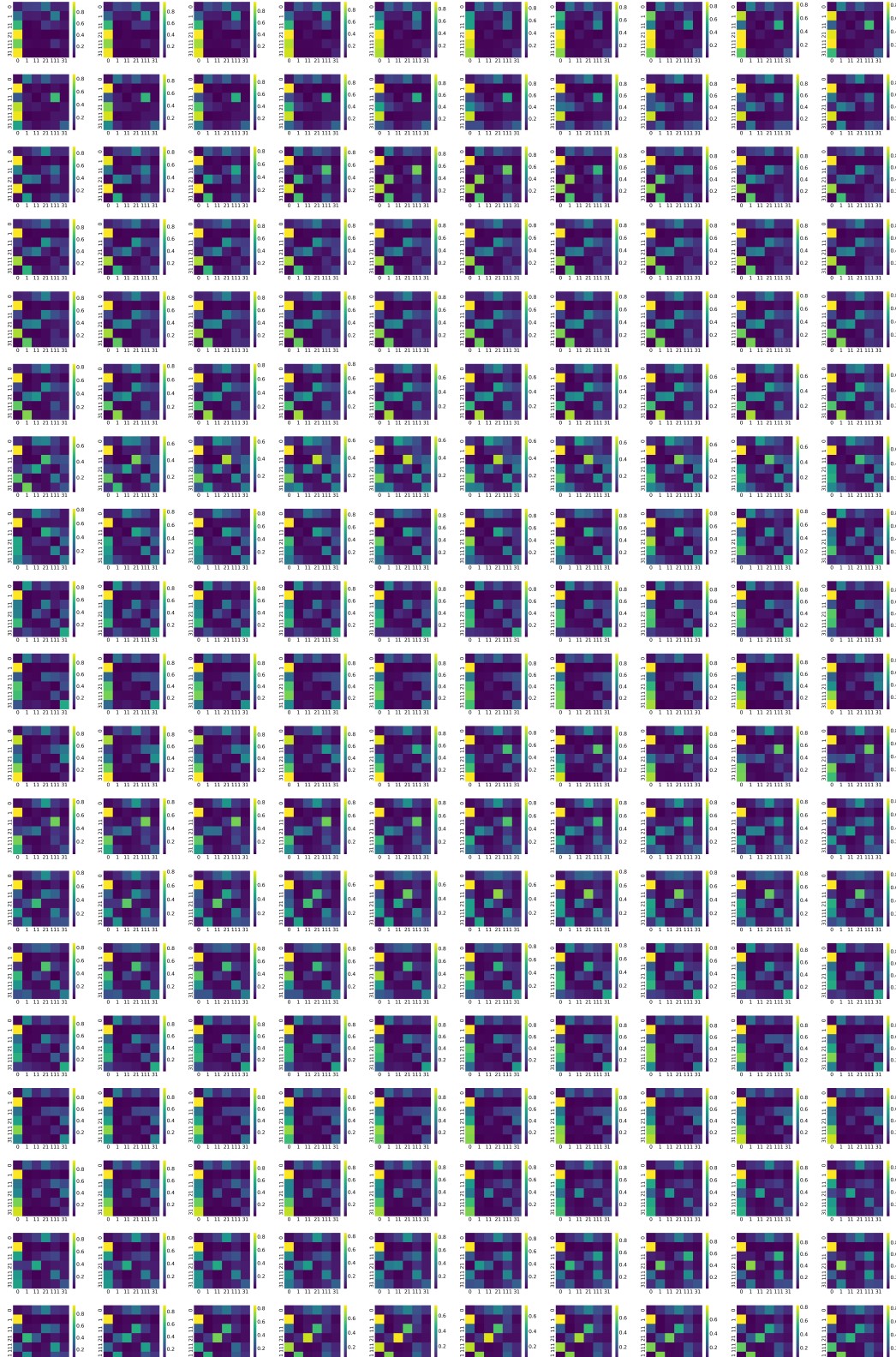

Figure 14: Visualization for BodyGen's attention map during the control process on Cheetah.

