# OpenReview forum: "BodyGen: Advancing Towards Efficient Embodiment Co-Design"
_ICLR.cc/2025/Conference — ICLR 2025 Spotlight_

### Official Review · Reviewer_GZKf · 2024-10-31

**Soundness:** 4
**Presentation:** 3
**Contribution:** 4
**Rating:** 8
**Confidence:** 4

**Summary:**

The paper presents "Genesis", a reinforcement learning (RL) framework for the co-design of robot morphology and control policies. Genesis addresses the main limitations of current methodologies which struggle with the high-dimensional design space and optimize the morphology separately from the control policy. The key contributions include (1) a temporal credit assignment mechanism to balance reward between morphology and control optimization and (2) a topology-aware self-attention mechanism for efficient morphology representation. Experiments across multiple environments and ablations demonstrate that Genesis significantly improves performance compared to baseline models.

**Strengths:**

- Genesis introduces an innovative end-to-end co-design optimization that combines attention-based morphology processing with reward balancing between morphology and control.
- The experiments demonstrate significant improvements over current methods, and the ablations highlight the importance of the topographical positional encoder.
- The paper generally explains its contributions and mechanisms clearly, with nice illustrations.
- This approach addresses important limitations in the field and, therefore, could influence future research in co-design optimization.

**Weaknesses:**

- Although Genesis demonstrates substantial improvements over current methods, the paper lacks a detailed explanation of specific solution differences between Genesis and previous approaches.
- Genesis's morphological decisions are not fully interpretable, and a detailed discussion of the morphological features it favors could enhance the paper.
- While Genesis aims for continuous co-optimization, this approach risks overfitting to specific scenarios or static environments.
- The clarity of some paragraphs related to the methods could be improved.

**Questions:**

- Does Genesis achieve better solutions than evolutionary search methods simply due to access to a superior hyperparameter space (e.g., varying thickness that may affect inertia and speed)? A direct comparison of best morphologies across methods, including an exploration of the role of hyperparameter configurations, would strengthen this aspect.
- Does Genesis prioritize certain features to boost robustness or performance in various environments? Furthermore, it would be good to visualize how morphologies evolve over time. Are the retrieved morphologies consistent across different seeds, or does Genesis find alternative solutions depending on initialization? Additionally, it would be interesting to analyze the topographical encoder representation and how attention shifts based on morphology.
- In real-world applications, slight morphological changes may occur. Ideally, the control policy should adapt to such changes in a zero-shot manner, as in [1,2,3]. A key question is how well Genesis handles IID morphologies (those seen in training) versus OOD morphologies.
- Once the best morphology has been found, does optimizing it independently yield the same or better performance? Does the control policy of Genesis benefit from the co-training?
- What are the main failure modes of Genesis?
- Why is model size evaluated in MB rather than by the number of parameters?
- What is U_t in equation 4.12?

[1]: Chiappa, A. S., Marin Vargas, A., & Mathis, A. (2022). Dmap: a distributed morphological attention policy for learning to locomote with a changing body. Advances in Neural Information Processing Systems.

[2]: Hong, S., Yoon, D., & Kim, K. E. (2021). Structure-aware transformer policy for inhomogeneous multi-task reinforcement learning. In International Conference on Learning Representations.

[3]: Huang, W., Mordatch, I., & Pathak, D. (2020). One policy to control them all: Shared modular policies for agent-agnostic control. In International Conference on Machine Learning.

---

> ### Author Response · Authors · 2024-11-22
> **Response to reviewer GZKf [1/2]**
>
> Thank you very much for your constructive comments and suggestions. We have revised our paper accordingly. Below, we will provide detailed responses to each point.
>
> > Q1: Does Genesis achieve better solutions than evolutionary search methods simply due to access to a superior hyperparameter space (e.g., varying thickness that may affect inertia and speed)? A direct comparison of best morphologies across methods, including an exploration of the role of hyperparameter configurations, would strengthen this aspect.
>
> Thanks for the suggestion. The design space of all the baselines is the same. The main reasons RL-based methods outperform evolutionary search methods are as follows:
>
> 1) ES methods do not allow experience sharing among species within a generation. For example, given $N$ interactions with the environment, each agent can only be trained with $N/P$ samples.
> 2) RL-based methods utilize a policy for morphology generation rather than relying on random mutation and filtering.
>
> > Q2: Does Genesis prioritize certain features to boost robustness or performance in various environments? Furthermore, it would be good to visualize how morphologies evolve over time. Are the retrieved morphologies consistent across different seeds, or does Genesis find alternative solutions depending on initialization? Additionally, it would be interesting to analyze the topographical encoder representation and how attention shifts based on morphology.
>
> We have updated more visualizations and attention maps in our Appendix. We will add more discussions in our camera ready version. Thank you for your suggestions.
>
> > Q3: In real-world applications, slight morphological changes may occur. Ideally, the control policy should adapt to such changes in a zero-shot manner, as in [1,2,3]. A key question is how well Genesis handles IID morphologies (those seen in training) versus OOD morphologies.
>
> Thank you for the question. We want to claim that the focus of Genesis and these papers are different.
>
> In [1], their topology is fixed, and morphology changes are passive and simulated through random perturbations, rather than purposefully evolving toward an optimal morphology. In [2] and [3], they are trying to learn a multi-task policy to control several morphologies at the same time. The focus of these papers are on learning an general or adaptive policy over a group of morphologies. Consequently, their OOD abilities requires evaluation. However, for Genesis, we aim to simultaneously optimize both morphology and policy towards the best one. At the end of training, the controlling policy is mainly suitable for the best corresponding morphology. Consequently, it may weaken its OOD abilities. Meanwhile, we add extensive Genesis's OOD ability with out-of-distribution environment settings on "crawler".
>
> | Setting Changes (%) | Gravity Coefficient        | Land Stiffness            | Land Damping             | Friction Coefficient     |
> |-|-|-|-|-|
> | -25.00%             | $5189.4 \pm 1439.4$       | $11310.0 \pm 54.6$        | $11320.8 \pm 12.5$       | $11232.3 \pm 88.1$       |
> | -20.00%             | $7916.0 \pm 372.1$        | $11263.6 \pm 59.2$        | $11118.9 \pm 123.0$      | $10804.7 \pm 260.5$      |
> | -15.00%             | $9480.9 \pm 225.3$        | $11152.4 \pm 184.7$       | $11249.2 \pm 85.4$       | $11282.3 \pm 40.3$       |
> | -10.00%             | $10709.9 \pm 168.3$       | $11248.6 \pm 92.6$        | $11358.3 \pm 66.5$       | $11300.5 \pm 73.5$       |
> | -5.00%              | $11248.7 \pm 11.4$        | $11345.6 \pm 71.2$        | $11446.3 \pm 20.7$       | $11266.2 \pm 37.4$       |
> | 0.00% (Original)    | $11179.5 \pm 38.8$        | $11179.5 \pm 38.8$        | $11179.5 \pm 38.8$       | $11179.5 \pm 38.8$       |
> | +5.00%              | $11240.1 \pm 152.4$       | $11365.5 \pm 69.9$        | $11386.7 \pm 61.7$       | $11142.0 \pm 50.7$       |
> | +10.00%             | $11229.7 \pm 65.9$        | $11232.5 \pm 99.6$        | $11324.0 \pm 71.6$       | $11281.2 \pm 39.5$       |
> | +15.00%             | $10847.4 \pm 36.8$        | $11231.5 \pm 71.9$        | $11322.7 \pm 41.4$       | $11089.0 \pm 98.6$       |
> | +20.00%             | $10453.2 \pm 58.5$        | $10796.3 \pm 331.0$       | $11274.6 \pm 34.0$       | $10995.1 \pm 52.1$       |
> | +25.00%             | $9405.3 \pm 129.4$        | $8907.9 \pm 1525.4$       | $11145.1 \pm 61.8$       | $10890.9 \pm 34.7$       |
> | +30.00%             | $8541.8 \pm 49.5$         | $6520.8 \pm 1490.9$       | $11222.3 \pm 160.1$      | $10536.5 \pm 49.6$       |
> | +40.00%             | $7486.2 \pm 26.7$         | $4892.4 \pm 1237.7$       | $11230.3 \pm 83.1$       | $9968.3 \pm 130.3$       |
>
> Although the goal of Genesis is to optimize the agent best suited to the current environment, experiments have shown that our approach can effectively generalize to out-of-distribution environments, with acceptable performance degradation in specific cases such as land stiffness (+40%).

---

> ### Author Response · Authors · 2024-11-22
> **Response to reviewer GZKf [2/2]**
>
> > Q4: Once the best morphology has been found, does optimizing it independently yield the same or better performance? Does the control policy of Genesis benefit from the co-training?
>
> Yes. Once the best morphology has been found, optimizing it independently yield the same or better performance, with even fewer environment interaction steps.
>
> For example, we have done experiments on the "Cheetah" environment using the same PPO and MoSAT settings. The learning curve is shown in [this picture](https://ibb.co/ZLCFmJg). After using Genesis to find the optimal morphology, training a control policy for this morphology from scratch with PPO requires fewer steps. This is reasonable, as directly optimizing a given morphology is much easier than embodiment co-design. At the same time, this also demonstrates that the morphologies generated by Genesis can efficiently be trained from scratch using MoSAT.
>
> > Q5: What are the main failure modes of Genesis?
>
> Thanks for the question. As shown in A.7 visualization, for the second Swimmer, some extra, non-functional small limbs appear. Similarly, in the challenging environment Walker-Hard ([video link](https://genesisorigin.github.io/static/videos/walker-hard.mp4)), additional limbs are observed near the body center. Occasionally, agents may develop irrelevant limbs that do not affect performance. Incorporating human guidance into the co-design process to influence the evolution of morphology and policy is an interesting research direction in the future.
>
> > Q6: What is U_t in equation 4.12?
>
> U_t is the discounted-free Monte-Calo return of the inner loop.
>
> > Q7: Why is model size evaluated in MB rather than by the number of parameters?
>
> Thank you for the advice. We provide parameter comparison with previous baselines here.
>
> We provide the total number of weights for all the baselines below:
>
> | Models           | Agent Parameters | Population Size           | Total Parameters     |
> |-------------------|------------------|---------------------------|------------|
> | Genesis (Ours)       | 1.43 M           | 1                         | 1.43 M     |
> | Transform2Act | 19.64 M          | 1                         | 19.64 M    |
> | UMC-Message*  | 0.27 M           | 1                         | 0.27 M     |
> | NGE           | 0.27 M          | 20 |  5.4 M   |
>
> *Note: For NGE, the total Models Required is $20 + 20 \times 0.15 \times 125 = 395$ (population_size + population_size * elimination_rate * generations). The total weights are derived using population_ size only.
>
> Thank you again for your constructive feedback, which we believe have significantly enhanced our work. If you have any further comments, we would be glad to discuss with you.

---

> > ### Comment · Reviewer_GZKf · 2024-11-25
> >
> > Thank you for thoroughly addressing all my concerns and questions, even considering the already high score I initially provided.
> >
> > - The final morphologies presented are intriguing. I’m curious if they exhibit any specific characteristics that confer particular performance advantages.
> > - I appreciate the demonstration of robustness in the developed morphology under OOD settings. I wonder, though, whether there is a measurable advantage in robustness between the evolved morphology and the initial one.

---

> ### Author Response · Authors · 2024-11-27
> **Re: Official Comment by Reviewer GZKf**
>
> We thanks for your constructive feedback.
>
> > Following Q1: The final morphologies presented are intriguing. I’m curious if they exhibit any specific characteristics that confer particular performance advantages.
>
> This is a open question. We provide visualizations on our [webpage](https://genesisorigin.github.io/) for better analysis. As shown in [this video](https://genesisorigin.github.io/static/videos/Teasorv3.mp4), we observe the following general characteristics in the high-performance morphologies generated by Genesis:
>
> 1) **Minimal Use of Limbs for Locomotion**
>    Despite having access to a full design space, agents optimized for high-speed motion tend to minimize the number of active limbs used for propulsion. This allows for more concentrated force generation while employing other body parts to maintain balance effectively.
>
> 2) **Flight-like Running Dynamics**
>    Compared to low-speed motion, morphologies designed by Genesis exhibit a tendency to adopt flight-like running patterns with increased aerial phases. For instance:
>    - In 3D environments ([e.g. this video](https://genesisorigin.github.io/static/videos/crawler.mp4)), agents evolve wing-like structures on lateral limbs to maintain balance during high-speed aerial phases.
>    - In 2D environments, some extreme agents ([e.g. this video](https://genesisorigin.github.io/static/videos/walker-regular.mp4)) opt to use a single limb for pendulum-like motion, reducing ground contact for other body parts.
>
> These characteristics highlight the ability of Genesis to evolve morphologies that are not only efficient but also adaptive to the task and environmental constraints. We also expect more future biological analysis based on Genesis.
>
> > Following Q2: I appreciate the demonstration of robustness in the developed morphology under OOD settings. I wonder, though, whether there is a measurable advantage in robustness between the evolved morphology and the initial one.
>
> Thank you for your question. We have to admit that it may be difficult to fairly compare the OOD ability of the initial designs with those generated by our approach. As shown in Figure 8, the initial designs consist of only 1–2 limbs of equal length, and some designs even lack effective locomotion capability (e.g., Type 3). This limitation makes direct comparisons challenging and may not provide a fair evaluation of OOD performance. We believe that evolving morphologies specifically for enhanced OOD capabilities in varying environments is an interesting research direction. Though it falls outside the scope of this paper, we look forward to seeing exciting work on this topic in the future.

---

### Official Review · Reviewer_Qsbs · 2024-10-31

**Soundness:** 4
**Presentation:** 3
**Contribution:** 2
**Rating:** 6
**Confidence:** 4

**Summary:**

This work tackles the problem of learning a control policy and improving the agent's morphology in a unified learning framework. The authors propose to improve both the morphology of the agent and the control policy via reinforcement learning.  Several previous works employed a combination of reinforcement learning for learning the policy and of evolutionary algorithms to improve the agent's embodiment, using a new neural network for each agent. Other papers used policies based on graph neural networks to enable weight sharing across morphologies, but suffered from inefficient communication between joints. With an attention-based policy, it is possible to both share the network's weights across different morphologies, and avoid the message passing bottleneck. The morphology self-attention architecture, paired with a mechanism to facilitate credit assignment for the policy improving the morphology, largely outperforms previous methods in co-evolving the agent's morphology and control policy.

**Strengths:**

The paper is well written and easy to follow. The motivation is clear: previous work employed certain design choices (specifically evolutionary algorithms, graph neural networks) which, if successfully replaced with more effective solutions (reinforcement learning and self-attention) should lead to a clear improvement, as indeed demonstrated by the final performance. The proposed framework required the introduction of new solutions for the positional encoding of the agent's joints and to improve the credit assignment for the "slower" reinfrocement learning process, which is the morphology improvement. Both problems have been successfully addressed. Therefore, the paper fills a a gap in this specific field, obtaining a large performance improvement over previous methods.

**Weaknesses:**

In the introduction and in the related work, this paper cites previous studies on policy-morphology co-evolution. However, the authors likely missed other papers proposing attention policies to deal with variable morphologies ([1] and [2]) and it would be important to at least compare MoSAT with those architectures. Furthermore, from the description of MoSAT I could not understand whether it is any different from a standard Transformer network. I would suggest explaining MoSAT in terms of its differences from a Transformer, rather than defining each component of the block.

The paper includes some details about methods introduced by other papers (e.g., details about PPO, or about the topology design, which seems identical to [3]), which could be shortened and the saved space used to better analyze the results. In fact, while the performance is convincing and the ablations satisfactory, the authors could draw inspiration from, e.g., the analysis in [4], to better grasp the effects of the morphology's evolution beyond the task performance.

The topology position encoding (TopoPE) is listed among the main contributions, so it would be good to better show its properties. For example, the authors claim that TopoPE is more stable than previous encodings when the morphology changes, but this claim is not supported by any specific experiment.

The modified GAE requires better explanation. For the "Control Stage", why is there a discounted advantage summed to the temporal difference? While for the Design Stage, U_t seems like a monte-carlo estimation of the reward, why using that instead of bootstrapping the value function?

Minor:
* line 080, facilitating -> facilitates
* line 139, an -> a
* line 242, "achieve the co-design process"?
* line 369, what exactly is inspired by BERT?
* line 443, Yuan et al. repeated
* line 463, I would rather express the size of the model in number of weights
* line 479, I would avoid sentences like "despite the impressive results of our approach"

[1] Trabucco, B., Phielipp, M., & Berseth, G. (2022, June). Anymorph: Learning transferable polices by inferring agent morphology. In International Conference on Machine Learning (pp. 21677-21691). PMLR.
[2] Chiappa, A. S., Marin Vargas, A., & Mathis, A. (2022). Dmap: a distributed morphological attention policy for learning to locomote with a changing body. Advances in Neural Information Processing Systems, 35, 37214-37227.
[3] Yuan, Ye, et al. "Transform2act: Learning a transform-and-control policy for efficient agent design." arXiv preprint arXiv:2110.03659 (2021).
[4] Gupta, A., Savarese, S., Ganguli, S. and Fei-Fei, L., 2021. Embodied intelligence via learning and evolution. Nature communications, 12(1), p.5721.

**Questions:**

* How does TopoPE compare to other morphology encodings? E.g., the one used in [1]

* Is the morphology search space the same as in [2] (unimal)? What is different and why?

* How large is the delay in the Design stage? It is not obvious to me that the rewards come particularly delayed, unless the design stage has many steps. In fact, while the reward due to the updated design comes after some interactions of the agent with the environment, during the interaction there is no design decision being made, so this should not be perceived as a delay by the Design policy.

* How do the agents' morphologies look at the end of the training? What is their performance, e.g., compared to agents with the base morphology trained with RL? Can the learnt morphology be used to train a new agent with it?

[1] Trabucco, B., Phielipp, M., & Berseth, G. (2022, June). Anymorph: Learning transferable polices by inferring agent morphology. In International Conference on Machine Learning (pp. 21677-21691). PMLR.
[2] Gupta, A., Savarese, S., Ganguli, S. and Fei-Fei, L., 2021. Embodied intelligence via learning and evolution. Nature communications, 12(1), p.5721.

---

> ### Author Response · Authors · 2024-11-22
> **Response to reviewer Qsbs [1/2]**
>
> Thank you very much for your constructive comments and suggestions. We have revised our paper accordingly. Below, we will provide detailed responses to each point.
>
> > Q1: How does TopoPE compare to other morphology encodings? E.g., the one used in [1]
>
> We apologize for the lack of clarity. First, we would like to clarify that most morphology encodings in previous works focus on Universal Morphology Control, where the policy can only handle a fixed number of morphology variants (typically 5-15) by randomly removing limbs from a full model. Representative approaches either use traversal sequences (e.g., DFS) or simply remove parts of the morphology. The method described in [1] can be summarized as follows: they perform a one-pass DFS on the full model and manually name each limb. When a limb is removed, the names of the remaining limbs remain unchanged.
>
> In contrast, in our task settings, there is no "full model," and we cannot name each limb manually. To address this, we propose "topology indexing," which uses the path to the root as the embedding index. This approach not only retains the advantageous properties of [1] but also facilitates alignment for agents with dynamic morphologies.
>
> > Q2: Is the morphology search space the same as in [2] (unimal)? What is different and why?
>
> Thank you for the question. In our work, Genesis includes 10 different tasks with various search spaces. The search space of Unimal is similar with our "crawler" environment. (Actually, the search space of unimal is smaller than "crawler"). We will compare them from several perspectives:
>
> **Initial Design:**
>
> The Unimal uses an "ant XML" with four limbs as the initial design: [Unimal Ant XML](https://github.com/agrimgupta92/derl/blob/main/derl/envs/assets/unimal_ant.xml), with one body and four limbs pointing in perpendicular directions.
> We use the same initial design: [Genesis Ant XML](https://github.com/GenesisOrigin/Genesis/blob/main/assets/mujoco_envs/ant.xml), also featuring one body and four limbs pointing in perpendicular directions.
>
> **Morphology Actions:**
>
> Unimal supports three types of morphology actions for random mutation:
> 1) Delete limb
> 2) Modify limb parameters
> 3) Add limbs
>
> Genesis supports three types of actions during the topology design stage and attribute design stage:
> 1) Add a limb  (Topo Design)
> 2) Delete a limb (Topo Design)
> 3) Pass (Topo Design)
> 4) Modify limb parameters (Attribute Design)
>
> The basic morphology actions are the same.
>
> **Search Space:**
>
> Unimal supports 10 max limbs (Table 2 in [2]'s Supplementary Material) while the "crawler" support up to 29 limbs.
>
> **Computational Demands:**
>
> Unimal uses 1152 CPUs for MuJoCo simulation and at least one GPU for policy training. In contrast, Genesis requires approximately 30 hours to train a model using 20 CPU cores and a single NVIDIA RTX 3090 GPU on our server.
>
> > Q3: How large is the delay in the Design stage? It is not obvious to me that the rewards come particularly delayed, unless the design stage has many steps. In fact, while the reward due to the updated design comes after some interactions of the agent with the environment, during the interaction there is no design decision being made, so this should not be perceived as a delay by the Design policy.
>
> We apologize for the lack of clarity. There are two types of delay: 1) The zero-rewarded design stage; 2) The discounted reward signals from the control stage. An action in the control stage has a decreasing impact on the future (one motion error can be corrected with future actions), whereas an action in the design stage should have an equal impact on every step in the control stage. Otherwise, the agent may focus disproportionately on the first few steps (e.g., evolving a body that makes the agent prone to falling early in an episode to exploit the reward brought by the fast "falling down" speed).

---

> > ### Comment · Reviewer_Qsbs · 2024-11-25
> >
> > Thank you for the clarification about the difference between TopoPE and the prior art, and also for the extension of the search space compared to Unimal. I would suggest to highlight these aspects in the text to highlight the contribution of this work. I also appreciated the videos, the learnt morphologies are interesting and somehow unexpected.
> >
> > The performance gain compared to the handcrafted morphology is clear and I think it is an important result that retraining the resulting morphology from scratch can be done without loss noticeable of performance compared to training the morphology together with the policy. This suggests that the morphology is, even by itself, a useful result of the co-evolution.
> >
> > Regarding the comparison with [1] and [2], it seems to me that in both works the attention is across limbs and not time. I agree with you that the setting is different, no evolution is performed in the prior work, but in my opinion these papers should be mentioned among the related works as examples of attention across body parts.
> >
> > *There are two types of delay: 1) The zero-rewarded design stage; 2) The discounted reward signals from the control stage. An action in the control stage has a decreasing impact on the future (one motion error can be corrected with future actions), whereas an action in the design stage should have an equal impact on every step in the control stage. Otherwise, the agent may focus disproportionately on the first few steps (e.g., evolving a body that makes the agent prone to falling early in an episode to exploit the reward brought by the fast "falling down" speed).*
> >
> > Here I agree with your explanation. Could you clarify, however, if at least the second problem (focusing disproportionally on the first steps) could as well be addressed by not discounting the trajectory reward for the design agent? After all, it does not take any action during the policy's rollouts, so it would be reasonable not to discount later rewards in the trajectory. I also asked a question about the GAE (The modified GAE requires better explanation. For the "Control Stage", why is there a discounted advantage summed to the temporal difference? While for the Design Stage, U_t seems like a monte-carlo estimation of the reward, why using that instead of bootstrapping the value function?), could you please provide some more information?
> >
> > *In this paper, we do not aim to create a highly specialized, expert-designed network for specific tasks. Instead, our goal is to showcase that a standard and general-purpose Transformer layer, combined with efficient morphology representation through our proposed TopoPE and enhanced TCA, is sufficient to unlock performance in robot co-design. This performance is often hindered by inefficient morphology representation and unbalanced reward signals (Figure 6, Figure 11, Figure 12).*
> >
> > This confirms what I understood from reading the paper, I would recommend making it more explicit that you are using a standard transformer block. The lines 263-289 describe a transformer block, but instead of simply writing that, you write "our model utilizes dot-product self-attention", then describe the key-query-value components, fully connected networks, skip connections... all components that are implemented for example in the PyTorch TransformerEncoder class. I think most readers will be confused by this part, thinking that something unusual is being described here, while it is the description of a transformer block.
> >
> > *We understand your concerns. We provide experiments in Table 1, Figure 11 to demonstrate the performance improvement brought by the topology position encoding.*
> >
> > Here I would like to clarify my question, I was not doubting that TopoPE leads to a performance improvement, as demonstrated by your experiments. I wanted a clarification about its stability properties. You show in fig. 4 that traversal PE can cause a complete change of the indexing even with a small morphological change, and you argue that this is not the case for TopoPE. If you are able to provide some theoretical reason to justify why this is the case, or even just showing experimentally that small changes to the morphology correspond to small changes to the encoding, your claim would be better supported.
> >
> > As many of my questions have been already answered, I have adjusted the score. I would still invite the authors to answer my remaining questions, while thanking them for the time they have already dedicated to the rebuttal.

---

> ### Author Response · Authors · 2024-11-22
> **Response to reviewer Qsbs [2/2]**
>
> > Q4: How do the agents' morphologies look at the end of the training? What is their performance, e.g., compared to agents with the base morphology trained with RL? Can the learnt morphology be used to train a new agent with it?
>
> Thank you for the question. We have provided visualizations of the embodied agents generated by Genesis in Appendix A.7. Additionally, we provide videos on our website: [https://genesisorigin.github.io/](https://genesisorigin.github.io/) for better visualizations.
>
> We trained the corresponding expert-designed robots (provided in Gym by OpenAI) using PPO, and compare their convergence performance with Genesis:
>
> | Initial Design | Handcrafted | Genesis  |
> |----------------|---------------------|----------|
> | Type#1         | 6869.3           | 10381.9 |
> | Type#2         | 130.4             | 1305.1  |
> | Type#3         | 3127.7           | 11611.5 |
> | Type#4         | 4895.6           | 11082.2 |
>
> We can observe significant performance improvement after embodiment co-design with Genesis.
>
> Yes. Given the final morphology (our framework supports exporting the final morphology in XML format), learning to control this morphology becomes a traditional control problem that can be addressed using RL algorithms. For example, we have done experiments on the "Cheetah" environment using the same PPO and MoSAT settings. The learning curve is shown in [this picture](https://ibb.co/ZLCFmJg). After using Genesis to find the optimal morphology, we can train a control policy for this morphology from scratch with PPO.
>
> > Q5: Comparison with attention policies in [1] and [2]
>
> Thanks for the suggestion. We have compared Genesis with [1] in Q1. Regarding [2], besides their fixed topology settings, in their work, morphology changes are passive and simulated through random perturbations, rather than purposefully evolving toward an optimal morphology. The focus of [2] is on learning an adaptive policy and performing attention in the time dimension. In contrast, our study takes an active approach to simultaneously optimize both morphology and policy, with the settings of topology changeability. Our attention mechanism operates at the limb level, simulating direct signal transmission within the body. The focus of these papers are on learning an general or adaptive policy over a group of morphologies, which is different from ours. We have added related discussions in our Appendix.
>
> > Q6: Question about the transformer layer in MoSAT.
>
> We apologize for the lack of clarity. MoSAT consists of attention modules and TopoPE, designed to achieve topology-aware self-attention. The attention module in MoSAT is implemented using a standard transformer layer.
>
> In this paper, we do not aim to create a highly specialized, expert-designed network for specific tasks. Instead, our goal is to showcase that a standard and general-purpose Transformer layer, combined with efficient morphology representation through our proposed TopoPE and enhanced TCA, is sufficient to unlock performance in robot co-design. This performance is often hindered by inefficient morphology representation and unbalanced reward signals (Figure 6, Figure 11, Figure 12).
>
> > Q7: The topology position encoding (TopoPE) is listed among the main contributions, so it would be good to better show its properties. For example, the authors claim that TopoPE is more stable than previous encodings when the morphology changes, but this claim is not supported by any specific experiment.
>
> We understand your concerns. We provide experiments in Table 1, Figure 11 to demonstrate the performance improvement brought by the topology position encoding.
>
> > Minors and writing suggestions
>
> Thank you for your careful reading. We have fixed these typos and revised our paper.
>
> [1] Trabucco, B., Phielipp, M., & Berseth, G. (2022, June). Anymorph: Learning transferable polices by inferring agent morphology. In International Conference on Machine Learning (pp. 21677-21691). PMLR.
>
> [2] Chiappa, A. S., Marin Vargas, A., & Mathis, A. (2022). Dmap: a distributed morphological attention policy for learning to locomote with a changing body. Advances in Neural Information Processing Systems, 35, 37214-37227.
>
> Finally, we would like to thank you again for the useful comments, which we believe have significantly enhanced our work. If you have any further comments, we would be glad to have a deeper discussion with you.

---

> ### Author Response · Authors · 2024-11-27
> **Re: Official Comment by Reviewer Qsbs [1/2]**
>
> Thank you very much for your insightful follow-up questions. Below, we provide detailed responses to each point:
>
> > **Following Q1**: I would suggest highlighting these aspects in the text to emphasize the contribution of this work.
>
> Thank you for the suggestion. We have revised our paper accordingly. Additional discussions have been included in Appendix A.4.2 and A.8, further elaborating on the contributions of Genesis.
>
> > **Following Q2**: Whether the second problem could as well be addressed by not discounting the trajectory reward for the design agent? After all ... it would be reasonable not to discount later rewards in the trajectory.
>
> > **Following Q3**: While for the Design Stage, $U_t$ seems like a Monte-Carlo estimation of the reward, why use that instead of bootstrapping the value function? Could you please provide more information?
>
> Your understanding is correct. That is precisely the approach taken in Genesis. For embodiment co-design, we do not use the same MDP for the design and control stages. Instead, the design MDP calculates $U_t$ independently as the discounted-free Monte-Carlo return of the inner loop.
>
> Since there are no direct rewards in the Design Stage, and we use the discounted-free Monte-Carlo return, bootstrapping the value function can be expressed as:
>
> $$
> U_t = r_t + U_{t+1} = U_{t+1}
> $$
>
> Thus, learning $V_\theta = r_t + \gamma V_\theta' = V_\theta'$ is equivalent to learning $V_\theta = U_t$. In practice, we find that the second formulation is more stable. This is because learning with a discounted-free return using extra TD-learning can lead to overestimation or underestimation issues. Empirical evidence supports the use of this approach for robustness and convergence.
>
> > **Following Q5**: I also asked a question about the GAE (The modified GAE requires better explanation. For the "Control Stage," why is there a discounted advantage summed to the temporal difference?)
>
> Thanks for the question. This factor, commonly known as the Generalized Advantage Estimator Factor (GAE Lambda: $\lambda$), was introduced by Schulman et al [1]. It uses an exponentially weighted average to estimate the advantage function, striking a balance between bias and variance. This technique improves both the stability and efficiency of policy gradient methods and is widely adopted in state-of-the-art implementations. Consequently, we maintain this coefficient in our implementation to align with classic practices in the RL community.
>
> > **Following Q6**: The lines 263-289 describe a transformer block, but instead of simply writing that, you write "our model utilizes dot-product self-attention", then describe the key-query-value components, fully connected networks, skip connections... all components that are implemented for example in the PyTorch TransformerEncoder class. I think most readers will be confused by this part, thinking that something unusual is being described here, while it is the description of a transformer block.
>
> Thanks for the suggestion. To align with the NLP community's usage of Transformers for processing one-dimensional sequences, we treat each limb’s information as a token and use TopoPE for indexing. This detailed explanation is intended to help readers better reproduce and understand our implementation. For more code details, researchers can refer to the following relevant code segments in our repository:
> - [Transformer implementation](https://github.com/GenesisOrigin/Genesis/blob/main/design_opt/models/transformer.py)
> - [Environment integration](https://github.com/GenesisOrigin/Genesis/blob/20df3ecac5549ef5e0f9f6a1d78addf5e1833f25/design_opt/envs/ant.py#L264-L270)
> - [Genesis policy logic](https://github.com/GenesisOrigin/Genesis/blob/20df3ecac5549ef5e0f9f6a1d78addf5e1833f25/design_opt/models/genesis_policy.py#L92-L152)
> - [Genesis value logic](https://github.com/GenesisOrigin/Genesis/blob/20df3ecac5549ef5e0f9f6a1d78addf5e1833f25/design_opt/models/genesis_critic.py#L55-L111)
>
> For your concerns, we have now highlighted in the main text that: "*MoSAT leverages a standard Transformer layer and our proposed TopoPE to achieve topology-aware self-attention across limbs*" in our revision version. We will further optimize and simplify our explanations in the camera-ready version to enhance clarity and accessibility. Thank you for your valuable feedback.
>
> [1] Schulman J, Moritz P, Levine S, et al. High-dimensional continuous control using generalized advantage estimation[J]. arXiv preprint arXiv:1506.02438, 2015.

---

> ### Author Response · Authors · 2024-11-27
> **Re: Official Comment by Reviewer Qsbs [2/2]**
>
> > **Following Q7**: TopoPE leads to a performance improvement... You show in Fig. 4 that traversal PE can cause a complete change of the indexing even with a small morphological change, and this is not the case for TopoPE. If you are able to provide some theoretical or experimental reason to justify why this is the case ... your claim would be better supported.
>
> Thank you for the question. We have provided our straightforward motivation in Figure 4. For morphologically dynamic structures, directly using DFS order for limb indexing can lead to indexing misalignment, which negatively impacts knowledge alignment and learning. Below, we further address this question from an experimental perspective:
>
> We claim that TopoPE can better adapt to changeable morphology structures in a reasonably alignable manner. To verify this, we conducted an experiment on the "crawler" environment, testing the embeddings when experiencing a newly "grown limb" at different positions, using a same learned control policy. For traversal position embedding, we first retrieved all the learned position embeddings and then re-indexed them using DFS from 1 to the end.
>
> | Position (in DFS) | Traversal PE| TopoPE   | TopoPE's Advantage |
> |--------------------|-----------|----------|----------------------|
> | 1                 | -0.9      | **1372.0**   | >100000.0%         |
> | 2                 | -25.6     | **145.6**    | >100000.0%         |
> | 3                 | 19.4      | **1264.3**   | +6423.7%           |
> | 4                 | 34.8      | **713.3**    | +1950.8%           |
> | 5                 | 98.5      | **940.3**    | +854.8%            |
> | 6                 | 44.7      | **11071.1**  | +24678.6%          |
> | 7                 | 12.0      | **9556.4**   | +79602.8%          |
> | 8                 | 66.0      | **6440.8**   | +9661.7%           |
> | 9                 | 233.5     | **956.4**    | +309.6%            |
> | 10                | **3552.7**    | 1056.1   | -70.3%             |
> | 11                | 1796.1    | **4037.4**   | +124.8%            |
> | 12                | 90.8      | **5148.5**   | +5568.3%           |
> | 13                | 1094.9    | **10836.0**  | +889.7%            |
> | 14                | 8437.5    | **11244.9**  | +33.3%             |
> | 15                | 7845.4    | **9665.2**   | +23.2%             |
> | 16                | 8744.7    | **10534.9**  | +20.5%             |
> | 17                | 9134.9    | **10951.5**  | +19.9%             |
> | 18                | 8787.7    | **10039.3**  | +14.2%             |
> | 19                | 8829.9    | **11108.2**  | +25.8%             |
> | 20                | **9855.8**    | 9767.7   | -0.9%              |
> | 21                | 10685.7   | **11181.8**  | +4.6%              |
> | 22                | 10841.3   | **11088.3**  | +2.3%              |
> | 23                | 11063.0   | **11253.8**  | +1.7%              |
> | 24                | **11293.0**   | 11086.6  | -1.8%              |
> | 25                | 10939.3   | **10936.2**  | +0.0%              |
> | 26                | 11150.8   | **11222.1**  | +0.6%              |
>
> As shown in the table, TopoPE adapts better to topology changes with an average lower performance influence when encounter morphology changes, compared with traversal PE. We believe the above experiment aligns with the explanations provided in Figure 4, and hope this could address your concern.
>
> Thank you for these valuable questions, which have allowed us to further clarify these aspects.

---

> > ### Comment · Reviewer_Qsbs · 2024-11-27
> >
> > Thank you once again for the clarifications, I have one last doubt. In several points in the paper you define TopoPE as a positional encoding. Why do you say you need to modify the transformer's implementation? Cannot you just add the encoding to the input tokens?
> >
> > I have briefly checked the code and I only see some non-standard logic in TransformerSimple, where one can choose among three different embeddings. But this does not influence the transformer block, so I don't really see why you cannot use the default transformer block. Also, the attribute pos_emb_type looks unused in MaskedSelfAttention and in the TransformerBlock is just passed to MaskedSelfAttention (therefore unused also in this case). So I maintain my impression that the operations are the ones of a standard transformer, where you add a custom positional encoding. Please correct me if I have misunderstood something, otherwise I encourage you to describe the architecture in these terms, which are much more easily understandable and quite standard in many other works also outside of the RL research (I mean transformer + ad-hoc positional encoder).

---

> ### Author Response · Authors · 2024-11-27
> **Re: Official Comment by Reviewer Qsbs**
>
> We sincerely appreciate your thoughtful review and constructive feedback. We are pleased to address your following comments:
>
> > Regarding the transformer operations in our implementation
>
> As detailed in Q6 and our main paper (Figure 3), we highlight that MoSAT utilizes a standard transformer layer for its attention module. For us, manually implementing the transformer block facilitates better in-batch padding and the integration of positional encoding, just for better coding. We agree that TopoPE's computation logic operates independently of the attention computation, making our approach compatible with various transformer implementations. We are grateful for your additional religious clarification.
>
> > Concerning the description of MoSAT in Section 4.1
>
> We acknowledge your valuable suggestion about improving clarity and conciseness. In response, we have streamlined the description of MoSAT (L254-L290) to enhance readability without sacrificing accuracy. The revised version, marked in blue, provides a simplified, more focused explanation. Additionally, we have added explicit clarification in L249, ensuring readers can quickly grasp the core concepts of our work.
>
> We believe these revisions have strengthened the paper while maintaining its technical rigor. If you find that we have addressed your concerns, we would greatly appreciate your reconsideration of the paper's ratings. We remain available to address any additional questions you may have.

---

### Official Review · Reviewer_yGDd · 2024-11-02

**Soundness:** 3
**Presentation:** 4
**Contribution:** 3
**Rating:** 8
**Confidence:** 3

**Summary:**

The paper introduces a robotics co-design framework that co-optimized morphology and control using a Morphology Self-attention (MoSAT) architecture, inspired by transformers. It employs Topology Position Encoding (TopoPE) to enhance the integration of positional data within self-attention mechanisms. The framework optimizes these configurations through a Proximal Policy Optimization (PPO) approach, incorporating specialized credit assignments to accommodate both immediate control actions and long-term design adjustments.

**Strengths:**

The authors propose a new method to overcome the computationally heavy nested design of morphology and control of robotics systems. Based on the literature review provided, the authors have substantially improved the drawbacks of previous approaches, and the results are sufficient to prove it. The proposed method significantly will alter future research in the co-design domain.

**Weaknesses:**

I expect a big computational increase due to the 3 complex neural networks involved. The paper didn't talk about it or compare it with baselines.
The robots and simulation environments used are toy problems and not sure how this can be applied to complex robotics systems like UGVs and UAVs.

**Questions:**

1. What are the computational demands of the MoSAT architecture, particularly when scaling to robots with a high number of limbs or complex joint structures? Is there a limit to the number of nodes (limbs) and edges (joints) the system can effectively manage?
2. How well does the Topology Position Encoding (TopoPE) generalize across robots with significantly different morphologies? Are there any limitations in the encoding approach when dealing with highly irregular or non-standard robotic structures?
3. given the use of multiple networks (Topology Design, Attribute Design, and Control) within the Genesis framework, how is the synchronization of updates managed between these networks? Let's say with episode I, with timestep t that is not a terminal step, if a backpropagation is performed here. How does the t+1 work, does the morphology change and controls change?

---

> ### Author Response · Authors · 2024-11-22
> **Response to reviewer yGDd**
>
> Thank you very much for your constructive comments and suggestions. We have revised our paper accordingly. Below, we will provide detailed responses to each point.
>
> > Q1: What are the computational demands of the MoSAT architecture, particularly when scaling to robots with a high number of limbs or complex joint structures? Is there a limit to the number of nodes (limbs) and edges (joints) the system can effectively manage?
>
> Thank you for the question. For all the environments used in our work, training a model takes approximately 30 hours on a server with 20 CPU cores and a single NVIDIA RTX 3090 GPU.
>
> Admittedly, an $\mathcal{O}(n^2)$ computation is required for cross-body attention when using a vanilla transformer layer (where $n$ is the number of limbs). However, there are increasingly mature solutions from the NLP community that reduce the quadratic computation complexity of attention, such as Linear Attention, RetNet, and State-Space Models. Our framework can easily integrate these solutions by replacing the transformer layers in MoSAT to reduce computation costs.
>
> Furthermore, the topology complexity for real-world robots, including complex humanoids, remains manageable with current hardware. For instance, the Unitree G1, a complex full humanoid robot, has 37 topology limbs with motors, sufficient to perform complex locomotion and manipulation tasks.
>
> We also manually pad the inputs to 100 joints, and the expected training time using Genesis is approximately 3 days, 8 hours, and 7 minutes on a single NVIDIA RTX 3090 GPU.
>
> > Q2: How well does the Topology Position Encoding (TopoPE) generalize across robots with significantly different morphologies? Are there any limitations in the encoding approach when dealing with highly irregular or non-standard robotic structures?
>
> This is a good question! For the implementation of TopoPE, we only experimented with learnable position embeddings in this paper. As a result, it is challenging to directly apply the learned TopoPE weights to another agent with significantly different morphology.
>
> To address this issue, a natural idea is to retain the indexing logic of our topology position embedding while using non-learnable position embeddings, such as `SinusoidalEmbedding`. These embeddings not only offer better extrapolation but also eliminate the need for training. We plan to include this option in future updates to our code repository.
>
> > Q3: Given the use of multiple networks (Topology Design, Attribute Design, and Control) within the Genesis framework, how is the synchronization of updates managed between these networks?
>
> We apologize for the lack of clarity. These networks are updated in a structured and synchronized manner. The gradient steps for Topology Design, Attribute Design, and Control Policies are identical. First, we collect all the transitions from an episode and mark the source of each action (Topology Net, Attribute Net, or Control Net). Then, we classify these transitions into three groups and perform one gradient step for each policy independently. The updates for the three policies are conducted separately.
>
> > Q4: I expect a big computational increase due to the 3 complex neural networks involved. The paper didn't talk about it or compare it with baselines.
>
> We provide the total number of weights for all the baselines below:
>
> | Models           | Agent Parameters | Population Size           | Total Parameters     |
> |-------------------|------------------|---------------------------|------------|
> | Genesis (Ours)       | 1.43 M           | 1                         | 1.43 M     |
> | Transform2Act | 19.64 M          | 1                         | 19.64 M    |
> | UMC-Message*  | 0.27 M           | 1                         | 0.27 M     |
> | NGE           | 0.27 M          | 20 |  5.4 M   |
>
> *Note: For NGE, the total Models Required is $20 + 20 \times 0.15 \times 125 = 395$ (population_size + population_size * elimination_rate * generations). The total weights are derived using population_ size only.
>
> Compared to previous works that use a specific MLP for each limb or rely on a population of agents, our network benefits from its lightweight model size and do not bring big computational increase. Based on our tests, it can perform inference on an M2 MacBook using only the CPU, achieving a frame rate of **90+** FPS. You are welcomed to give it a try with [interactive visualization section](https://github.com/GenesisOrigin/Genesis?tab=readme-ov-file#interactive-visualization) in our repository.
>
> > Q5: The robots and simulation environments used are toy problems and not sure how this can be applied to complex robotics systems like UGVs and UAVs.
>
> We understand your concerns. As discussed in Q1, Genesis demonstrates potential for real-world robotic applications. However, we also acknowledge that research in this area is still in its early stages. We hope our framework can contribute to its further development. Thank you for the question again.

---

### Official Review · Reviewer_JKmz · 2024-11-07

**Soundness:** 3
**Presentation:** 3
**Contribution:** 3
**Rating:** 8
**Confidence:** 3

**Summary:**

The authors investigate the problem of joint optimization of agent design and control with reinforcement learning. They propose that two outstanding problems for addressing this are imbalanced reward signals between the control and design generation steps and the effective use of morphology information. They address these two problems with two components. First, they address this by modifying the advantage function to give a better signal during design development. The latter is addressed by proposing the MoSAT layer, which uses self-attention instead of a graph neural network and an embodiment-aware positional encoding approach. Experimentally the combination of these components is necessary for performance benefits based on their results

**Strengths:**

Overall, the paper proposes several sound components that address their proposed problems (reward mismatch and using morphology information). Modifying the training objective for the design policy makes intuitive sense, at leas twith how the author's describe it. Self-attention mechanisms have been effective in other embodiment-aware research, so using these mechanisms seems a natural step for co-design and control methods. The authors also compare their positional embedding mechanism to prior works, which adds support to the benefit of their embedding approach. The experiments used to validate the components, and the author provides supporting materials support their claims

**Weaknesses:**

Our biggest concern is that although the proposed framework yields strong performance, the authors need to pay more attention to the novelty of their contributions. We point out, for example, that the author's "Topology Design Stage" sounds similar to the previous work in Transform2Act (Yuan et al., 2021, in their citations). The MoSAT layer is simply a typical Transformer Encoder layer, so it seems strange that the authors renamed a previously established neural network model. Stating this would simplify their discussion on the MoSAT layers and allow more room for discussing other components. The reward signal modifications do not adequately provide readers with the intuition of the author's proposed solution in section 4.3. In our estimates, the major contribution is merging these components into a single design and control deployment framework, which is a good contribution. Still, the components in themselves do not seem notably novel alone or need to be justified more clearly to this reviewer.


Writing comments:
- Replace the statement "remains a tough nut to crack" with a more representative statement.
- Consider citing MAT: Morphological Adaptive Transformer for Universal Morphology Policy Learning Li et al 2024 as related work for morphology-aware positional encoding. They focus only on the control aspects.
 - In Ablation experiments, grouping the Figure 6 ablations might be better than discussing the Table 1 ablations separately. "Figure 6 includes ablations of X & Y, where Table 1 shows ablations of Positional differences".

**Questions:**

1. What distinguishes the author's Topology Design Stage from the design stage in Transform2Act? They also seem to have a graph-generating component and attribute-generating policy.

2. How is MoSAT different from a typical Transformer Layer?

3. What exactly is U_t in comparison to the value function V?

4. How was 60.52% performance improvement calculated?

5. In the w/o TopoPE ablation, was any form of positional information provided? Also, is the adjacency information included explicitly in the limb embeddings?

---

> ### Author Response · Authors · 2024-11-22
> **Response to reviewer JKmz**
>
> Thank you very much for your constructive comments and suggestions. We have revised our paper accordingly. Below, we will provide detailed responses to each point.
>
> > Q1: What distinguishes the author's Topology Design Stage from the design stage in Transform2Act? They also seem to have a graph-generating component and attribute-generating policy.
>
> Thank you for the question. The aim of the "Topology Design Stage" is the same as the skeleton transform stage in Transform2Act: to generate the agent's body topology.
>
> In practical implementation, there are differences. (1) Genesis generates a robot's topology in an autoregressive manner during this stage. The topology generated at the current step is fed to MoSAT in the next step, where new limbs are registered to pick up their Topology Position Embedding. (2) In Transform2Act, the topology is modified by adding nodes and updating a maintained adjacency matrix. We retain similar stage functionality to enable better performance evaluations across different baselines.
>
> > Q2: How is MoSAT different from a typical Transformer Layer?
>
> MoSAT consists of attention modules and TopoPE, designed to achieve topology-aware self-attention. The attention module in MoSAT is implemented using a standard transformer layer.
>
> In this paper, we do not aim to design a highly specialized, expert-designed network for specific tasks. Instead, our goal is to showcase that a standard and general-purpose Transformer layer, combined with efficient morphology representation through our proposed TopoPE and enhanced TCA, is sufficient to unlock performance in robot co-design. (Figure 6, Figure 11, Figure 12).
>
> To the best of our knowledge, this work may be the first step to leverage transformers to generate a robot's morphology in a GPT-like autoregressive manner and control the entire body with the same architecture. We believe this approach better aligns with practices in the language and vision communities. Conseqently, we name the framework as MoSAT.
>
> > Q3: What exactly is U_t in comparison to the value function V?
>
> U_t is the discounted-free Monte-Calo return of the inner loop.
>
> > Q4: How was 60.52% performance improvement calculated?
>
> We apologize for the lack of clarity. The average performance improvement compared to the best baseline on each task is calculated as follows:
>
> $\text{score} = \frac{1}{N} \sum_{\text{task}_i} \frac{\text{ours} - \max{\{\text{baselines on task}_i\}}}{\max{\{\text{baselines on task}_i\}}}$
>
> The baselines include {NGE, Transform2Act, UMC-Message, ours w/o MoSAT, and ours w/o Enhanced TCA} as listed in Table 3.
>
> The previously reported rate of 60.52% was calculated using an earlier version. The updated result is 60.03%, reflecting minor adjustments due to an increased number of seeds for robust testing. The correction has been made—thank you for bringing this to our attention. For easy reproduction of all experiments, we also provide our code in PyTorch, which is available at [https://github.com/GenesisOrigin/Genesis](https://github.com/GenesisOrigin/Genesis).
>
> > Q5: In the w/o TopoPE ablation, was any form of positional information provided?
> > Q6: Also, is the adjacency information included explicitly in the limb embeddings?
>
> We apologize for the lack of clarity
> 1) No. But there's an one-hot input in each limb's observation-"limb depth", describing the distance from the root.
> 2) The adjacency information is not included explicitly.
>
> > Writing Coments:
>
> Thank you for your suggestions and careful reading.
> 1) We have replaced the statement "remains a tough nut to crack" with "remains complex and challenging to address".
> 2) We have include MAT: Morphological Adaptive Transformer for Universal Morphology Policy Learning (Li et al 2024) as related work for morphology-aware positional encoding.
> 3) We have added "Figure 6 presents the ablation studies for TopoPE and Enhanced-TCA, while Table 1 highlights the differences for different positional embedding choices."
>
> Finally, we would like to thank you again for the useful comments, which we believe have significantly enhanced our work. If you have any further comments, we would be glad to have a deeper discussion with you.

---

> > ### Comment · Reviewer_JKmz · 2024-11-25
> > **Explanation for score increase - plus one comment**
> >
> > We chose to increase our score because the authors adequately clarified our concerns.  Our only other comment is the authors is to better distinguish their work from that of [1]. This prior work also uses a Transformer model for joint design and control, so it would be good to discuss the critical differences between the author's work and this previous work.
> >
> > We mention this point because of the following comment in the author's response:
> >   "To the best of our knowledge, this work may be the first step to leverage transformers to generate a robot's morphology in a GPT-like autoregressive manner and control the entire body with the same architecture".
> >
> > We strongly encourage the authors to re-evaluate whether the above statement is true about how transformers are used to [1].
> >
> > [1] Wang, Y., et al. "PreCo: Enhancing Generalization in Co-Design of Modular Soft Robots via Brain-Body Pre-Training." Conference on Robot Learning. PMLR, 2023.

---

> ### Author Response · Authors · 2024-11-27
> **Re: Explanation for score increase - plus one comment**
>
> > Following: This prior work [1] also uses a Transformer model for joint design and control, so it would be good to discuss the critical differences between the author's work and this previous work.
>
> Thank you for your question. We believe our claim is valid and we want to address your concern from two key aspects:
>
> **#1 Differences Between Rigid Robots and Modular Soft Robots (MSRs)**
>
> In [1], the authors focus on Modular Soft Robots (MSRs) that consist of deformable cubes placed on a fixed-size grid (e.g., 5x5). These robots rely on a predefined "full model" (the NxN grid), where each voxel can be designated as either occupied (with 3-4 different types of materials) or empty. Consequently, they avoid topology misalignment issues because the grid inherently provides a simplified, structured full index space: a two-dimensional grid.
>
> In contrast, our task settings target real-world rigid robots, where no "full model" exists. This better aligns with general robot settings. Morphologies are generated dynamically, and limbs cannot be pre-named or indexed based on a predefined structure. To address this challenge, we propose "topology indexing," a method that enables consistent and efficient representation of agnostic and dynamically evolving morphologies.
>
> **#2 Autoregressive Design Using Transformer**
>
> In [1], robot generation actually occurs only in one step (Table 2 of their Appendix: Design Steps=1). The network is fed with the required inputs, and outputs the entire robot in a single forward pass, which is not an autoregressive process. In contrast, Genesis builds rigid robots limb-by-limb in an autoregressive manner using transformers. At each step, the generated topology is fed back into the system to condition the generation of the next limb. This iterative process reflects a more biologically plausible and modular approach to robot design and enables better handling of complex, real-world morphologies.
>
> We hope this clarifies the differences and strengthens the rationale for our approach. Thank you again for your acknowledge of our work.
>
> [1] Wang, Y., et al. "PreCo: Enhancing Generalization in Co-Design of Modular Soft Robots via Brain-Body Pre-Training." Conference on Robot Learning. PMLR, 2023.

---

### Author Response · Authors · 2024-11-22
**Response to all reviewers**

Thank you for the constructive feedback and the acknowledge of our work. We have revised our paper according to suggestions and we hope we have addressed your raised concerns.

The main changes in the manuscript are highlighted in blue, and here is a summary of our rebuttal:

1. **Typographical and Language Improvements**:
   - Corrected typos and improved the clarity of several statements throughout the paper.
   - Replaced ambiguous or informal phrases with precise language for better readability.

2. **Additional Discussions and Visualizations**:
   - Expanded the discussion on Morphology Position Embedding (TopoPE) in Appendix A.4.2, providing more insights into its implementation and advantages.
   - Included additional visualization results in Appendix A.7 to better illustrate the agents’ evolved morphologies and behaviors.

3. **Methodological Clarifications**:
   - Enhanced descriptions of critical components like MoSAT and Topology Design Stage.
   - Provided detailed explanations for experimental setups, evaluation metrics, and baseline comparisons, enhancing easy reproducibility.

We believe these revisions have significantly strengthened the manuscript. Please see the specific responses to individual reviewers for detailed explanations and discussions.

Thank you again for your thoughtful review, which has helped us improve the quality and clarity of our work. Please feel free to share any additional comments on the manuscript or the changes.

---

> ### Author Response · Authors · 2024-11-25
> **Follow-Up on Reviewer Comments**
>
> Dear reviewers, as the discussion period is nearing its conclusion, we wanted to take a moment to thank you again for your thoughtful feedback, which has greatly helped us improve our work. We hope our responses have addressed your comments effectively, and are happy to clarify any remaining points if needed.

---

### Meta-Review · Area_Chair_aurj · 2024-12-18

**Metareview:**

The paper presents an approach for morphology - policy codesign. The reviewers have rated this work quite highly and I agree, the paper presents several novel ideas along with strong execution and results. The strongest concern in my opinion is the large-scale applicability of this work. For example, could these ideas be used to optimize real-world robots?

**Additional Comments On Reviewer Discussion:**

Reviewers have increased the score after detailed responses from the authors.

---

### Decision · Program_Chairs · 2025-01-22

Accept (Spotlight)